# Defining and Discovering Hyper-meta-paths for Heterogeneous Hypergraphs

**Yaming Yang**[1], **Ziyu Zheng**[1], **Weigang Lu**[2], **Zhe Wang**[1], **Xinyan Huang**[3], **Wei Zhao**[1], **Ziyu Guan**[1]*

[1]School of Computer Science and Technology, Xidian University, China
[2]Department of Civil and Environmental Engineering,
The Hong Kong University of Science and Technology, Hong Kong, China
[3]Key Laboratory of Intelligent Perception and Image Understanding of Ministry of Education,
School of Artificial Intelligence, Xidian University, China
{yym@, zhengziyu@stu., wglu@stu., zwang@stu., ywzhao@mail., zyguan@}xidian.edu.cn

## Abstract

Heterogeneous hypergraph is a kind of structural data that contains multiple types of nodes and multiple types of hyperedges. Each hyperedge type corresponds to a specific multi-ary relation (called hyper-relation) among subsets of nodes, which goes beyond traditional pair-wise relations in simple graphs. Existing representation learning methods for heterogeneous hypergraphs typically learn embeddings for nodes and hyperedges based on graph neural networks. Although achieving promising performance, they are still limited in capturing more complex structural features and richer semantics conveyed by the composition of various hyper-relations. To fill this research gap, in this work, we propose the concept of hyper-meta-path for heterogeneous hypergraphs, which is defined as the composition of a sequence of hyper-relations. Besides, we design an attention-based heterogeneous hypergraph neural network (HHNN) to automatically learn the importance of hyper-meta-paths. By exploiting useful ones, HHNN is able to capture more complex structural features to boost the model's performance, as well as leverage their conveyed semantics to improve the model's interpretability. Extensive experiments show that HHNN can achieve significantly better performance than state-of-the-art baselines, and the discovered hyper-meta-paths bring good interpretability for the model predictions. To facilitate the reproducibility of this work, we provide our dataset as well as source code at: `https://github.com/zhengziyu77/HHNN`.

## 1 Introduction

Graph data is a common type of non-independent identically (non-i.i.d.) distributed data in our real-world life, which can be used to describe various complex relations between objects. Representation learning and knowledge mining on graph data are beneficial for many real-world scenarios, such as social networks [29], bioinformatics [42], e-commerce platform [43], and relational databases [13], etc. In this paper, we categorize the existing graph data into four categories based on their complexity, as shown in Figure 1.

❶ **Homogeneous Simple Graphs** consist of one type of nodes and one type of edges, and each edge represents a binary (pair-wise) relation between two nodes. Figure 1(a) shows a toy homogeneous simple graph, which describes five users and their pairwise friendship relations.

❷ **Heterogeneous Simple Graphs** consist of multiple types of nodes and multiple types of edges, with edges representing various types of binary relations between node pairs. Figure 1(b) shows a toy

---

*Corresponding author

39th Conference on Neural Information Processing Systems (NeurIPS 2025).

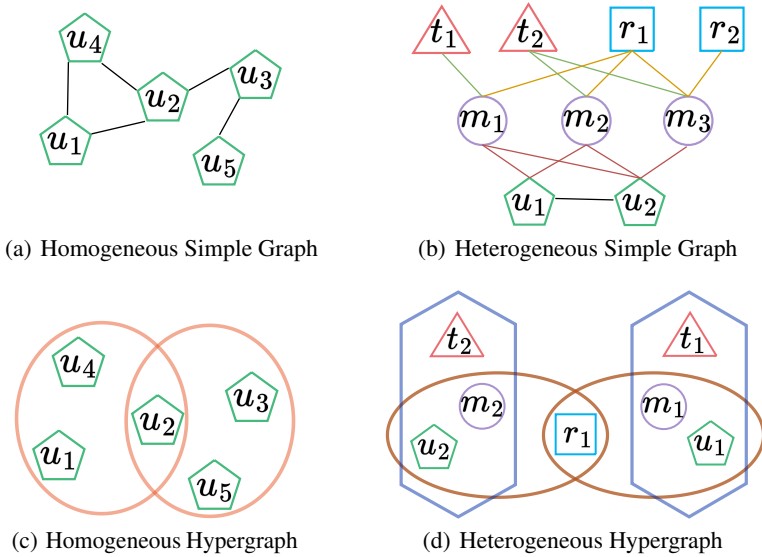

(a) Homogeneous Simple Graph      (b) Heterogeneous Simple Graph

(c) Homogeneous Hypergraph      (d) Heterogeneous Hypergraph

Figure 1: The toy examples of four categories of graph data.

heterogeneous simple graph, which includes four types of nodes: users, movies, tags, and ratings, denoted by different shapes, and four types of binary relations, denoted by different colors of lines.

❸ **Homogeneous Hypergraphs** generalize the binary relations between node pairs to multi-ary (tuple-wise) relations among a set of nodes. The multi-ary relation among a set of nodes is called a hyperedge. In this work, we also refer to the hyperedge type (i.e., the multi-ary relation) as the *hyper-relation*. Similar to homogeneous simple graphs, homogeneous hypergraphs consist of one type of nodes and one type of hyperedges. Figure 1(c) shows a toy homogeneous hypergraph, which describes a social group (i.e., a hyperedge) among users $u_1$, $u_2$, and $u_4$, and another social group among users $u_2$, $u_3$, and $u_4$.

❹ **Heterogeneous Hypergraphs** consist of multiple types of nodes and multiple types of hyperedges (hyper-relations), which can describe multiple types of hyper-relations among multiple types of nodes. Figure 1(d) shows a toy heterogeneous hypergraph, which contains four types of nodes (indicated by different shapes) and two types of hyperedges (indicated by different shapes with different colors). The two hyperedges $\{u_2, m_2, t_2\}$ and $\{u_1, m_1, t_1\}$ belong to the first type of hyper-relation, indicating the semantics that "a user associates a movie with a tag". The two hyperedges $\{u_2, m_2, r_1\}$ and $\{u_1, m_1, r_1\}$ belong to the second type of hyper-relation, indicating the semantics that "a user associates a movie with a rating".

We can observe that for heterogeneous hypergraphs, when the cardinality of their hyperedges is two, or the types of their nodes and hyperedges are one, they can be reduced to the other three categories of graphs, i.e., the latter are three special cases of the former. *In this work, we focus on the representation learning of heterogeneous hypergraphs (more than two types of nodes and more than two types of hyperedges) as they are not only the most general but also the most challenging types of graph data.*

At present, several heterogeneous hypergraph representation learning methods [4, 22, 26, 30, 10, 19, 25, 14] have been proposed. They are all graph neural network (GNN)-based methods, and follow the typical framework of GNNs by first aggregating information from nodes to hyperedges and then aggregating information from hyperedges back to nodes. They have achieved promising performance and have been successfully applied to practical scenarios such as social recommendation [22, 26], spatiotemporal activity prediction [25], and complex system analysis [11]. However, these methods fail to capture more complex structural features and richer semantics conveyed by various types of hyperedges. For the example shown in Figure 1(d), the two hyperedge types convey two different semantics as we have explained in Paragraph ❹. In addition, the two hyperedge types interact with each other through the shared node types (users and movies, i.e., the overlapping area of blue circles and brown circles), resulting in more complex semantics.

In this work, we aim to capture the rich semantics contained in heterogeneous hypergraphs. To this end, let us first review an important concept in heterogeneous simple graphs, i.e., meta-path, which is defined as a sequence of binary relations between nodes [31]. Referring to the example shown in Figure 1(b), let us use "$R_1$" to denote the binary relation of "users watch movies", and use "$R_2$" to denote the binary relation of "movies have tags". Then, we can compose the metapath "$R_1 \diamond R_2$", where "$\diamond$" denotes the composition operator. This meta-path describes the new composite binary relation between users and tags: "users watch movies that have specific tags". Meta-paths are often represented by node types as well. Here, the meta-path can also be denoted as $U \xrightarrow{R_1} M \xrightarrow{R_2} T$, where $U$, $M$, and $T$ denote the node types of users, movies, and tags, respectively. It is noteworthy that this meta-path is very different from the hyper-relation $\{U, M, T\}$, since from the meta-path, we cannot know who assigned these tags to the movie, while the hyper-relation conveys a more precise semantics of "users associate movies with specific tags". Meta-path has been proven to be very useful in capturing heterogeneous structural features from heterogeneous simple graphs [31, 8, 35, 40]. Unfortunately, it can only handle heterogeneous *simple graphs* and cannot be directly applied to heterogeneous *hypergraphs* due to the challenge of the multi-arity of hyper-relations.

To fill this research gap, we innovatively propose a novel concept called *hyper-meta-path*, which is formally defined as a sequence of hyper-relations, in which each pair of adjacent hyper-relations shares some common node types. It describes the composite relation of these hyper-relations. In this view, hyper-meta-path is defined in the spirit of the meta-path, with the former being a generalization of the latter. Hence, hyper-meta-path can capture more complex structural features and richer semantic information in heterogeneous hypergraphs.

Further, we develop a GNN-based representation learning model called Heterogeneous Hypergraph Neural Network (**HHNN**), which is able to automatically learn the importance of hyper-meta-paths. Each model layer goes through three levels of attention aggregation blocks to refine the representations of nodes and hyperedges. Finally, useful hyper-meta-paths can be discovered based on the optimized attention distributions. The merits are twofold. On the one hand, these hyper-meta-paths help the model capture more complex and subtle high-order structural features, boosting the performance of downstream analysis tasks. On the other hand, the rich semantics conveyed by these hyper-meta-paths endow the model with self-interpretability. That is, these semantics can bring some interesting interpretations to the model's predictions.

The main contributions of this work are summarized as follows:

- We are the first to define the concept of hyper-meta-path for heterogeneous hypergraphs. It describes the composite relation among a sequence of hyper-relations, capturing more complex structural features and richer semantics.

- We design a novel network architecture called Heterogeneous Hypergraph Neural Network (HHNN), which can automatically learn the importance of hyper-meta-paths. By discovering and exploiting useful hyper-meta-paths, HHNN is able to achieve higher performance as well as better self-interpretability.

- We conduct extensive experiments to study the effectiveness of the proposed concept of hyper-meta-paths and the proposed HHNN model. It turns out that HHNN can achieve significantly better performance than state-of-the-art baselines, and the discovered hyper-meta-paths bring good interpretability for the model predictions.

## 2 Related Work

**Homogeneous Hypergraph Neural Networks**. Inspired by the graph convolutional network (GCN) [24] for homogeneous simple graphs, HGNN [12] is the first work to define the spectral convolution on hypergraphs based on the hypergraph Laplacian, effectively extending GCN to homogeneous hypergraphs. HyperGCN [38] approximates each hyperedge by a set of pairwise edges that connect the nodes in the hyperedge. Inspired by the graph attention network (GAT) [33] for homogeneous simple graphs, Hyper-SAGNN [41] develops a self-attention based on the hypergraph neural network to deal with hypergraphs with variable hyperedge size. Inspired by the GraphSAGE model [17] for homogeneous simple graphs, HyperSAGE [2] proposes an inductive hypergraph learning framework that can capture the intra-relations (within a hyperedge) as well as inter-relations (across hyperedges). HNHN [7] introduces a generalized normalization aggregation scheme of GNN,

which weights the contributions of nodes/hyperedges by a power of their degrees, depending on a real-valued parameter. UniGNN [21] is proposed as a unified hypergraph learning framework, which generalizes several classic GNNs, such as GCN [24], GAT [33], GIN [37], and GraphSAGE [17] to hypergraphs. AllSet [6] implements hypergraph neural network layers as compositions of two multiset functions. ED-HNN [34] is a hypergraph neural network that can approximate any continuous equivariant hypergraph diffusion operators, and thus it can model a wide range of higher-order relations. These methods can effectively address the high-order structural features of hypergraphs, but they have limitations in capturing the heterogeneity of heterogeneous hypergraphs, since there may be multiple types of nodes as well as multiple types of hyperedges.

**Heterogeneous Hypergraph Neural Networks**. At present, there exists a series of heterogeneous hypergraph neural network methods [4, 30, 14, 25, 26, 10, 22, 19, 5, 18, 36], and the representative ones are introduced as follows. HHNE [4] is a GCN-based hypergraph neural network that can consider the heterogeneity of nodes by projecting different types of node features into a common feature space. HWNN [30] transforms heterogeneous simple graphs into a series of hypergraph snapshots based on a set of user-specified meta-paths. Then it conducts hypergraph convolution based on the Wavelet basis. HGNN+ [14] extends HGNN [12] to heterogeneous hypergraphs by introducing four ways to generate different types of hyperedges. DisenHCN [25] disentangles the user representations into different aspects (location-aware, time-aware, and activity-aware) and aggregates corresponding aspects' features on the constructed hypergraph. These methods not only can capture high-order structural features of hypergraphs but also can address the heterogeneity of nodes and hyperedges by various strategies. However, as we analyzed previously, they fail to capture the complex structural features and rich semantics conveyed by hyper-meta-paths.

## 3 Preliminaries

In this section, we first define some basic notations and concepts about heterogeneous hypergraphs. Then, we give the formal definition of our proposed concept of hyper-network-schema and hyper-meta-path, as illustrated by Figure 2.

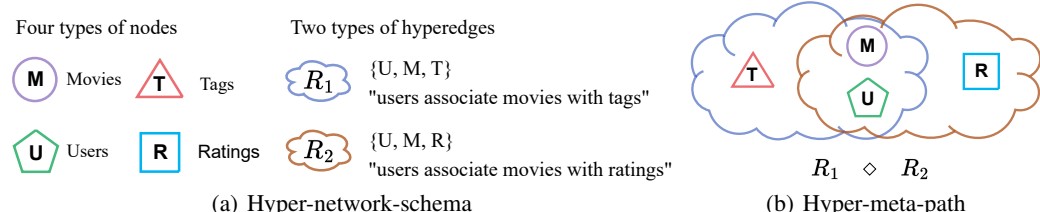

Figure 2: The illustration of hyper-network-schema and hyper-meta-path of the toy heterogeneous hypergraph shown in Figure 1(d). Note that we use "R" to denote node type "Ratings", and use "$R_x$" to denote hyperedge type $x$, i.e., hyper-relation $x$.

**Heterogeneous Hypergraphs.** A hypergraph is defined as $\mathcal{G} = (\mathcal{V}, \mathcal{E}, \mathbf{H}, \mathbf{W}, \phi, \psi)$, where $\mathcal{V}$ is a set of nodes, $\mathcal{E}$ is a set of hyperedges, and $\mathbf{H} \in \{0, 1\}^{|\mathcal{V}| \times |\mathcal{E}|}$ is an incidence matrix that describes the belonging relations between node and hyperedges, with the entry $\mathbf{H}_{i,j} = 1$ indicating that node $i$ is in the hyperedge $j$. The diagonal matrix $\mathbf{W} \in \mathbb{R}^{|\mathcal{E}| \times |\mathcal{E}|}$ stores the weights of hyperedges. $\phi$: $\mathcal{V} \rightarrow \mathcal{A}$ is a node type mapping function, and $\psi$: $\mathcal{E} \rightarrow \mathcal{R}$ is a hyperedge type mapping function, with $|\mathcal{A}| + |\mathcal{R}| > 2$. Let $\mathbf{v}_i$ and $\mathbf{e}_j$ denote the representation vectors of node $i$ and hyperedge $j$, respectively. Each hyperedge type $R_k \in \mathcal{R}$ conveys a specific multi-ary relation among all the associated, and thus we also call it hyper-relation $R_k$.

**Hyper-network-schema.** We define the concept of hyper-network-schema as a tuple $< \mathcal{A}, \mathcal{R} >$, describing the types of the nodes and the types of the hyperedges in a hypergraph, respectively.

A hypergraph $\mathcal{G}$ is homogeneous when $|\mathcal{A}| = 1$ as well as $|\mathcal{R}| = 1$. *In this work, we particularly focus on a more challenging setting by considering heterogeneous hypergraphs with $|\mathcal{A}| > 1$ as well as $|\mathcal{R}| > 1$, i.e., both the nodes and the hyperedges are heterogeneous.*

In Figure 2(a), we show the hyper-network-schema of the toy heterogeneous hypergraph that is previously shown in Figure 1(d). We can clearly see that there are four types of nodes, and two types of hyperedges, i.e., two hyper-relations in the heterogeneous hypergraph.

**Hyper-meta-path.** For each hyper-relation $R_k$ (the hyperedge type is $k$), we can denote it as its associated node types, i.e., $R_k = \{\phi(i)|\mathbf{H}_{i,j} = 1, \psi(j) = k, \forall i \in \mathcal{V}, \forall j \in \mathcal{E}, k \in \mathcal{R}\}$ to denote all the associated node types. Then, a hyper-meta-path with length $l$ is defined as a sequence of hyper-relations: $\mathcal{P} = R_1 \diamond R_2 \diamond \cdots \diamond R_l$, where $\diamond$ denotes the composition operator between hyper-relations, and two hyper-relations $R_x$ and $R_y$ can be composed if and only if: $R_x \cap R_y \neq \emptyset$.

In Figure 2(b), we intuitively show a hyper-meta-path $R_1 \diamond R_2$ according to the hyper-network-schema shown in Figure 2(a). As we can see, the hyper-meta-path describes a more complex hyper-relation by compositing the two hyper-relations $R_1$ and $R_2$, since they share the same node types, i.e., $R_1 \cap R_2 = \{M, U\}$. Besides, according to the semantics of $R_1$ and $R_2$, the hyper-meta-path conveys a richer semantics of "users associate specific movies with specific tags and specific ratings".

## 4 Methodology

The aggregation scheme of HHNN is shown in Figure 3. Firstly, it projects node features into a hyperedge-specific feature space, and leverages the $\alpha$-Attention to aggregate features from nodes to hyperedges. Then, it projects hyperedge features back to node-specific feature space, and leverages the $\beta$-Attention and $\gamma$-Attention to respectively perform intra-hyperedge-type aggregation and inter-hyperedge-type aggregation, resulting in the updated node features.

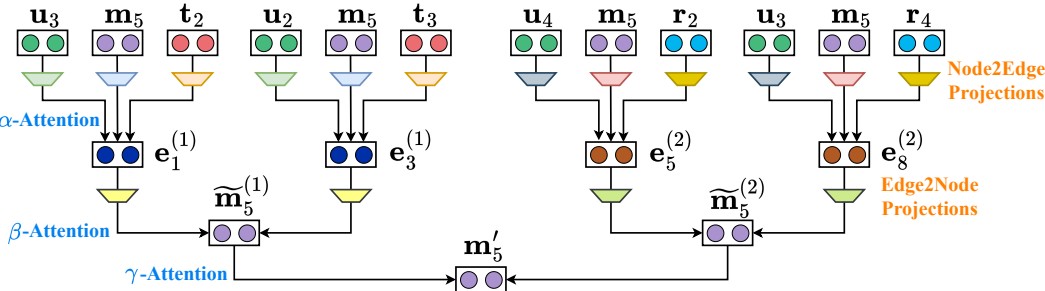

Figure 3: The aggregation scheme of HHNN.

### 4.1 Aggregation from Nodes to Hyperedges

Each hyperedge contains multiple nodes, and we define a set to describe all the nodes contained in hyperedge $j$:

$$\mathcal{N}_j^{\mathcal{V}} = \{i|\mathbf{H}_{i,j} = 1, i \in \mathcal{V}, j \in \mathcal{E}\} \tag{1}$$

In a heterogeneous hypergraph, the nodes in $\mathcal{N}_j^{\mathcal{V}}$ may have various types. To aggregate the features of these nodes, we need to first project them into a common feature space through parameter matrices, and these projection parameter matrices are specific to both the involved node type $\phi(i)$ and hyperedge type $\psi(j)$. Then, we leverage the attention mechanism to aggregate the projected node features together, resulting in the updated feature of hyperedge $j$, which is formally described as follows:

$$\mathbf{e}_j = \sum_{i \in \mathcal{N}_j^{\mathcal{V}}} \alpha_{j,i} \cdot \mathbf{W}^{\psi(j) \leftarrow \phi(i)} \cdot \mathbf{v}_i \tag{2}$$

where $\mathbf{W}^{\psi(j) \leftarrow \phi(i)}$ is a learnable parameter matrix and its superscript "$\psi(j) \leftarrow \phi(i)$" denotes projecting features from nodes of type $\phi(i)$ to hyperedges of type $\psi(j)$, $\alpha_{j,i}$ is the attention aggregation weight from node $i$ to hyperedge $j$, which is described as follows:

$$\alpha_{j,:} = \mathbf{Attention}_{\theta_{\psi(j)}}\left(\left\{\mathbf{W}^{\psi(j) \leftarrow \phi(i)} \cdot \mathbf{v}_i\right\}_{i \in \mathcal{N}_j^{\mathcal{V}}}\right) \tag{3}$$

where $\theta_{\psi(j)}$ contains the learnable parameters of the attention function, which are specific to hyperedge type $\psi(j)$ and serve as the attention query. The projected node representations can be mapped into the attention keys and attention values based on different projection parameter matrices.

## 4.2 Aggregation from Hyperedges to Nodes

In a heterogeneous hypergraph, a node may belong to multiple hyperedges of various types. For the toy example in Figure 1(d), node $m_2$ belongs to one hyperedge with "blue" type, and one hyperedge with "brown" type. Node $r_1$ belongs to two hyperedges with "brown" type. To this end, in the hyperedge-to-node aggregation phase, different from most previous approaches, we decompose the aggregation process into two levels, i.e., *intra-type aggregation* and *inter-type aggregation*. In the following, we describe the procedures by taking node $i$ as an example.

### 4.2.1 Intra-type Aggregation

For node $i$, assuming that it participates in multiple hyperedges of type $k$, we define a set to describe these hyperedges:

$$\mathcal{N}_i^{\mathcal{E}_k} = \{j | \mathbf{H}_{i,j} = 1, i \in \mathcal{V}, j \in \mathcal{E}, \psi(j) = k\} \tag{4}$$

Then, we leverage the feature projection and the attention mechanism again to aggregate the features of the hyperedges in $\mathcal{N}_i^{\mathcal{E}_k}$, resulting in a temporary representation for node $i$:

$$\widetilde{\mathbf{v}}_i^k = \sum_{j \in \mathcal{N}_i^{\mathcal{E}_k}} \beta_{i,j} \cdot \mathbf{W}^{\phi(i) \leftarrow \psi(j)} \cdot \mathbf{e}_j \tag{5}$$

where $\mathbf{W}^{\phi(i) \leftarrow \psi(j)}$ is a projection parameter matrix, and the superscript "$\phi(i) \leftarrow \psi(j)$" denotes projecting features from hyperedges of type $\psi(j)$ to nodes of type $\phi(i)$. $\beta_{i,j}$ is the attention aggregation weight from hyperedge $j$ to node $i$, which is calculated as follows:

$$\beta_{i,:} = \mathbf{Attention}_{\theta_{\phi(i)}} \left( \left\{ \mathbf{W}^{\phi(i) \leftarrow \psi(j)} \cdot \mathbf{e}_j \right\}_{j \in \mathcal{N}_i^{\mathcal{E}_k}} \right) \tag{6}$$

where $\theta_{\phi(i)}$ are the attention parameters specific to node type $\phi(i)$. Like the previous attention function, the attention parameters $\theta_{\phi(i)}$ serve as the attention query, and the inputs of the function, i.e., the projected hyperedge features, are mapped into attention keys and values.

Similar aggregation processes are conducted for all the possible hyperedge types that node $i$ participates in. Finally, we can obtain a set of temporary representations for node $i$, denoted as $\left\{ \widetilde{\mathbf{v}}_i^k | \mathcal{N}_i^{\mathcal{E}_k} \neq \emptyset, \forall k \in \mathcal{R} \right\}$, and abbreviated as $\left\{ \widetilde{\mathbf{v}}_i^k \right\}$. Its element $\mathbf{v}_i^k$ reflects the property of node $i$ from the aspect of hyper-relation $R_k$.

### 4.2.2 Inter-type Aggregation

To obtain a more comprehensive representation for node $i$, we use the attention mechanism once more to fuse the temporary representations of node $i$ from all the involved hyper-relations:

$$\mathbf{v}_i' = \sum_k \gamma_{i,k} \cdot \widetilde{\mathbf{v}}_i^k \tag{7}$$

where $\gamma_{i,k}, \forall k$ are the attention coefficients that are computed by an attention function as follows:

$$\gamma_{i,:} = \mathbf{Attention}_{\omega_{\phi(i)}} \left( \left\{ \widetilde{\mathbf{v}}_i^k \right\} \right) \tag{8}$$

where $\omega_{\phi(i)}$ are the attention parameters specific to node type $\phi(i)$, and serve as the attention query. The inputs of the attention function, i.e., the obtained temporary node representations, are mapped into the attention keys and values.

In this way, we conduct this three-level attention aggregation in each model layer to update the node representations and the hyperedge representations. Figure 4 shows the overall architecture of HHNN. We can obtain the final node embeddings $\{\mathbf{v}_i | \forall i \in \mathcal{V}\}$, and the final hyperedge embeddings $\{\mathbf{e}_j | \forall j \in \mathcal{E}\}$ from the last layer.

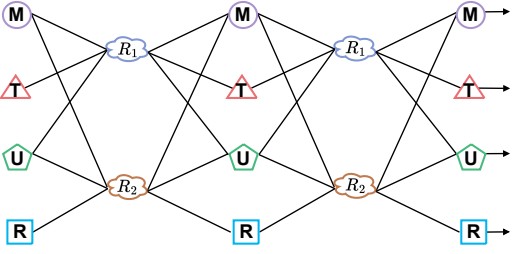

Figure 4: The architecture of HHNN.

### 4.3 Training Loss

By obtaining the embeddings of nodes and hyperedges, we can implement the loss function according to the task at hand. For example, for the semi-supervised node classification task, we utilize the cross-entropy loss as follows:

$$L = -\sum_{i \in \mathcal{Y}} \mathbf{y}_i \cdot \ln(\mathbf{C} \cdot \mathbf{v}_i) \tag{9}$$

where $\mathbf{v}_i$ and $\mathbf{y}_i$ denote the extracted embedding and the ground-truth label of node $i$, respectively, $\mathcal{Y}$ denotes the set of node indices that have labels, and $\mathbf{C}$ denotes the parameter matrix of the classifier.

In Appendix D, we show the overall algorithm of HHNN. In Appendix E, we theoretically show that HHNN can automatically discover the most important hyper-meta-path for each node, which helps interpret the model's prediction. In Section 5.3, we verify this ability of HHNN through case studies.

### 4.4 Time Complexity Analysis

The time complexity of HHNN is analyzed as follows. For each layer, the main cost comes from the three-level attention aggregation operations. The $\alpha$-attention aggregation involves projecting and attending to nodes within each hyperedge, costing $\mathcal{O}(|E| \cdot d_\alpha \cdot f_\alpha \cdot r)$, where $|E|$ is the number of hyperedges, $d_\alpha$ and $f_\alpha$ are the dimensionalities of the input and output features, respectively, and $r$ denotes the average number of nodes per hyperedge. In the $\beta$-attention aggregation phase, each node aggregates its connected hyperedges of each type, costing $\mathcal{O}(|V| \cdot d_\beta \cdot f_\beta \cdot t)$ where $|V|$ is the number of nodes, $d_\beta$ and $f_\beta$ are the feature dimensionalities, and $t$ denotes the average number of hyperedges of each type that a node participates in. In the $\gamma$-attention aggregation phase, each node fuses its representations across different hyperedge types, costing $\mathcal{O}(|V| \cdot d_\gamma \cdot f_\gamma \cdot k)$, where $d_\gamma$ and $f_\gamma$ are the feature dimensionalities, and $k$ is the average number of hyperedge types. Assuming the model has $n$ layers, the total time complexity becomes: $\mathcal{O}(|E| \cdot d_\alpha \cdot f_\alpha \cdot r \cdot n + |V| \cdot d_\beta \cdot f_\beta \cdot t \cdot n + |V| \cdot d_\gamma \cdot f_\gamma \cdot k \cdot n)$. In practical scenarios, $d_\alpha, f_\alpha, d_\beta, f_\beta, d_\gamma, f_\gamma, r, t, k, n$ are usually much smaller constants compared to $|\mathcal{V}|$ and $|\mathcal{E}|$. Therefore, the total time complexity of HHNN is equal to: $\mathcal{O}(|\mathcal{V}| + |\mathcal{E}|)$, which is linear to the size of the hypergraph.

## 5 Experiment

In this section, we conduct comprehensive experiments on two real-world datasets. Please see Appendix A for dataset details, see Appendix B for the details of the used baseline methods, and see Appendix C for the detailed experimental settings.

### 5.1 Performance Comparison

Following most previous related studies [12, 14, 38, 7, 6, 21, 34], we compare the performance of all the methods by conducting node classification for the target nodes that are associated with ground-truth labels. The final experimental results are reported in Table 1 and Table 2.

As we can see, our proposed HHNN shows the best performance in all cases. This is due to the fact that HHNN is able to capture more complex and higher-order structural features that are conveyed by the discovered useful hyper-meta-paths. For all the methods, the performance is significantly better when the training ratio is higher, indicating that they can effectively exploit the supervision information. HGNN+ significantly outperforms HGNN in all cases because the former addresses hyperedge heterogeneity in a more nuanced manner. AllSetTsfm performs slightly better than AllDeepSets in most cases, which is also consistent with the similar finding reported in their original paper [6]. For most baselines as well as our HHNN, the experimental results on the Olist dataset are better than those on the Movielens dataset, which is because the target (Product) nodes in Olist are easier to distinguish, and the structural features conveyed by the three types of hyperedges are more informative.

### 5.2 Ablation Study

As described by Section 4, the key process of HHNN lies in three levels of attention aggregation blocks. As illustrated in the left part of Figure 2(a)), the $\alpha$-Attention block performs node-level

Table 1: Performance comparison on Movielens.

| Metrics | Micro | | Macro | | AUC | |
|---|---|---|---|---|---|---|
| Training Ratio | 20% | 40% | 20% | 40% | 20% | 40% |
| CE-GCN | $55.46_{\pm0.96}$ | $57.31_{\pm0.97}$ | $21.99_{\pm4.32}$ | $26.53_{\pm1.66}$ | $67.86_{\pm2.38}$ | $71.26_{\pm1.68}$ |
| HNHN | $52.53_{\pm0.79}$ | $54.42_{\pm1.33}$ | $15.67_{\pm1.92}$ | $17.36_{\pm3.00}$ | $64.61_{\pm3.14}$ | $67.99_{\pm2.72}$ |
| HyperGCN | $48.53_{\pm1.97}$ | $49.83_{\pm1.45}$ | $16.55_{\pm1.48}$ | $18.05_{\pm2.27}$ | $57.89_{\pm1.06}$ | $59.87_{\pm1.96}$ |
| AllDeepSets | $52.39_{\pm1.13}$ | $55.46_{\pm1.67}$ | $15.72_{\pm2.13}$ | $23.13_{\pm1.69}$ | $65.47_{\pm1.71}$ | $70.94_{\pm1.68}$ |
| AllSetTsfm | $53.39_{\pm1.84}$ | $57.40_{\pm1.60}$ | $18.01_{\pm5.52}$ | $30.32_{\pm3.13}$ | $66.21_{\pm2.97}$ | $73.63_{\pm2.03}$ |
| UniGNN | $52.35_{\pm1.08}$ | $54.18_{\pm1.58}$ | $16.41_{\pm2.42}$ | $24.11_{\pm3.43}$ | $65.78_{\pm2.16}$ | $71.77_{\pm1.39}$ |
| EDHNN | $53.41_{\pm0.60}$ | $56.72_{\pm1.08}$ | $18.74_{\pm2.60}$ | $32.56_{\pm3.22}$ | $70.07_{\pm1.31}$ | $77.06_{\pm1.15}$ |
| HGNN | $52.54_{\pm1.79}$ | $55.30_{\pm1.69}$ | $20.48_{\pm3.31}$ | $25.98_{\pm4.17}$ | $65.25_{\pm2.50}$ | $70.06_{\pm2.21}$ |
| HGNN+ | $53.13_{\pm2.21}$ | $55.44_{\pm1.76}$ | $21.62_{\pm3.21}$ | $26.13_{\pm2.70}$ | $69.62_{\pm2.60}$ | $73.77_{\pm1.62}$ |
| **HHNN** | $\mathbf{56.80}_{\pm1.65}$ | $\mathbf{60.65}_{\pm1.88}$ | $\mathbf{26.24}_{\pm3.90}$ | $\mathbf{33.33}_{\pm3.19}$ | $\mathbf{73.21}_{\pm2.13}$ | $\mathbf{78.72}_{\pm1.60}$ |

Table 2: Performance comparison on Olist.

| Metrics | Micro | | Macro | | AUC | |
|---|---|---|---|---|---|---|
| Training Ratio | 20% | 40% | 20% | 40% | 20% | 40% |
| CE-GCN | $32.31_{\pm0.70}$ | $42.97_{\pm0.62}$ | $31.29_{\pm0.74}$ | $42.12_{\pm0.63}$ | $70.20_{\pm0.30}$ | $77.76_{\pm0.21}$ |
| HNHN | $53.96_{\pm0.88}$ | $60.39_{\pm0.89}$ | $53.40_{\pm1.30}$ | $59.90_{\pm1.02}$ | $84.71_{\pm0.48}$ | $88.05_{\pm0.49}$ |
| HyperGCN | $21.96_{\pm0.52}$ | $27.96_{\pm0.79}$ | $20.21_{\pm0.64}$ | $25.71_{\pm1.33}$ | $62.08_{\pm0.62}$ | $67.66_{\pm1.05}$ |
| AllDeepSets | $66.96_{\pm0.49}$ | $75.89_{\pm0.79}$ | $66.95_{\pm0.46}$ | $75.89_{\pm0.80}$ | $90.29_{\pm0.30}$ | $94.44_{\pm0.24}$ |
| AllSetTsfm | $64.57_{\pm1.15}$ | $75.97_{\pm0.68}$ | $64.82_{\pm1.18}$ | $76.11_{\pm0.76}$ | $88.79_{\pm0.53}$ | $94.31_{\pm0.25}$ |
| UniGNN | $57.47_{\pm0.46}$ | $68.11_{\pm0.65}$ | $57.13_{\pm0.47}$ | $67.89_{\pm0.62}$ | $87.20_{\pm0.28}$ | $92.36_{\pm0.28}$ |
| EDHNN | $61.62_{\pm0.70}$ | $70.88_{\pm1.04}$ | $61.35_{\pm0.68}$ | $70.71_{\pm0.99}$ | $89.36_{\pm0.36}$ | $93.47_{\pm0.35}$ |
| HGNN | $40.66_{\pm0.61}$ | $48.62_{\pm0.58}$ | $40.11_{\pm0.60}$ | $48.38_{\pm0.62}$ | $76.19_{\pm0.51}$ | $80.93_{\pm0.31}$ |
| HGNN+ | $59.03_{\pm0.80}$ | $64.16_{\pm0.74}$ | $59.43_{\pm0.74}$ | $64.63_{\pm0.75}$ | $86.84_{\pm0.33}$ | $89.86_{\pm0.24}$ |
| **HHNN** | $\mathbf{72.54}_{\pm0.54}$ | $\mathbf{77.74}_{\pm0.83}$ | $\mathbf{72.89}_{\pm0.56}$ | $\mathbf{77.90}_{\pm0.80}$ | $\mathbf{93.20}_{\pm0.32}$ | $\mathbf{95.39}_{\pm0.25}$ |

aggregation to update hyperedge representations. The $\beta$-Attention block and $\gamma$-Attention block correspond to intra-type hyperedge aggregation and inter-type hyperedge aggregation, respectively. Here, we conduct comprehensive ablation studies to investigate the effectiveness of these three attention blocks. The experimental results are shown in Table 3. We use the binary bit "0" or "1" to indicate whether the corresponding attention blocks are activated. When an attention block is deactivated, we replace the attention aggregation with the mean aggregation. By considering all possible combinations of the three blocks, we obtain eight variants of HHNN. In the first column of Table 3, we also use three-bit binary numbers to denote different variants.

As we can see, in all the cases, the variant "111" achieves the best performance, which corresponds to the full version of our HHNN. This indicates that the power of HHNN can only be fully unleashed when all three attention blocks are assembled together, which implies that these three attention blocks can enhance each other. Besides, variant "010" performs better than variants "001" and "100", and variant "101" performs worse than variants "011" and "110", indicating that the $\beta$-Attention block is very effective. This indicates that the different hyperedges a node belongs to have significantly different levels of importance for it.

## 5.3 Hyper-meta-path Discovery and Interpretability Study

One of the merits of our HHNN is that it can discover useful hyper-meta-paths for the concerned task (see Appendix E). Here, we show this ability of HHNN by analyzing the learned attention coefficients after finishing the model training. In Figure 5(a) and Figure 5(b), we show the learned attention weights that are associated with the movie node $m_{2934}$ on Movielens, and the product node

Table 3: The performance of different variants of our HHNN.

| Variants Modes | Movielens | | | Olist | | |
|---|---|---|---|---|---|---|
| | Micro | Macro | AUC | Micro | Macro | AUC |
| 000 | $55.37_{\pm1.81}$ | $27.87_{\pm4.63}$ | $74.09_{\pm1.78}$ | $71.79_{\pm0.87}$ | $71.79_{\pm0.81}$ | $93.33_{\pm0.25}$ |
| 001 | $56.81_{\pm1.70}$ | $30.63_{\pm5.41}$ | $75.60_{\pm1.61}$ | $68.44_{\pm0.69}$ | $68.72_{\pm0.67}$ | $92.17_{\pm0.34}$ |
| 010 | $58.97_{\pm1.73}$ | $34.21_{\pm4.60}$ | $78.44_{\pm1.74}$ | $76.96_{\pm0.66}$ | $77.03_{\pm0.66}$ | $95.38_{\pm0.18}$ |
| 011 | $60.29_{\pm1.42}$ | $35.45_{\pm4.78}$ | $79.28_{\pm1.58}$ | $74.74_{\pm0.63}$ | $74.89_{\pm0.62}$ | $94.87_{\pm0.22}$ |
| 100 | $57.34_{\pm2.66}$ | $30.55_{\pm5.14}$ | $75.32_{\pm1.75}$ | $63.60_{\pm0.90}$ | $64.02_{\pm0.79}$ | $90.58_{\pm0.36}$ |
| 101 | $59.26_{\pm2.40}$ | $35.45_{\pm3.04}$ | $77.87_{\pm1.88}$ | $71.19_{\pm0.96}$ | $71.38_{\pm0.91}$ | $93.24_{\pm0.42}$ |
| 110 | $61.42_{\pm1.63}$ | $36.22_{\pm4.79}$ | $80.12_{\pm2.24}$ | $77.28_{\pm0.61}$ | $77.41_{\pm0.66}$ | $95.73_{\pm0.18}$ |
| **111** | $\mathbf{62.95}_{\pm1.94}$ | $\mathbf{39.03}_{\pm2.32}$ | $\mathbf{81.87}_{\pm1.85}$ | $\mathbf{79.82}_{\pm0.71}$ | $\mathbf{79.91}_{\pm0.78}$ | $\mathbf{96.32}_{\pm0.21}$ |

$p_{4593}$ on Olist, respectively. In the figures, the magnitude of the attention weights is reflected by the thickness of the corresponding arrows, and the notations in the figures are described in Table 4.

In Figure 5(a), for the target movie node, the most important path can be denoted as "T $\xleftarrow{R_1}$ U $\xleftarrow{R_2}$ M". Referring to the definition in Section 3, in the path, the hyper-relation $R_1$ (i.e., hyperedge type {UMT}) and the hyper-relation $R_2$ (i.e., hyperedge type {UMR}) are composited based on the movie node shared between them, forming the hyper-meta-path "$R_1 \diamond R_2$". Observing the path again, hyper-meta-path "$R_1 \diamond R_2$" connects the leftmost tag node to the rightmost movie (i.e., target) node. The full semantics conveyed by "$R_1 \diamond R_2$" can be described as: "the leftmost tag node participates in a {UMT} hyperedge" and "the rightmost movie node participates in a {UMR} hyperedge" and "the two hyperedges share the same user node", which can serve as the interpretation for the prediction result of the target movie node.

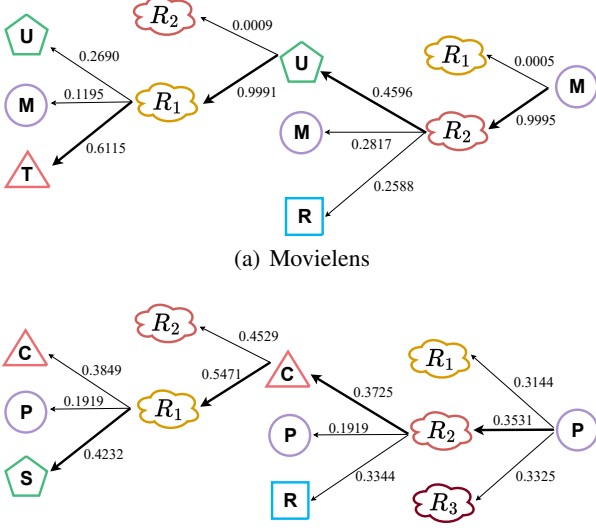

(a) Movielens

(b) Olist

Figure 5: Discovered hyper-meta-paths by HHNN.

Similarly, in Figure 5(b), the discovered most important hyper-meta-path can be denoted as "S $\xleftarrow{R_1}$ C $\xleftarrow{R_2}$ P". In the path, the hyper-relation $R_1$ (i.e., hyperedge type {CPS}) and the hyper-relation $R_2$ (i.e., hyperedge type {CPR}) are composited based on the customer node shared between them, forming the hyper-meta-path "$R_1 \diamond R_2$". The semantics of the hyper-meta-path can be described as: "the leftmost seller node participates in a {CPS} hyperedge" and "the rightmost product node participates in a {CPR} hyperedge" and "the two hyperedges share the same customer node". This semantics can help interpret the prediction result of the target product node.

## 5.4 Analysis of Different Hyperedges

We investigate the impact of the number of hyperedge types on HHNN performance. In Figure 6(a) and Figure 6(d), we show the performance of HHNN by exploiting each single type of hyperedges, as well as the performance by exploiting all types of hyperedges (marked by "ALL" in the legend).

As we can see, on both datasets, HHNN achieves the best performance when exploiting all types of hyperedges, which is due to the fact that HHNN can well handle the heterogeneity of hyperedges in an adaptive way, and it can compose more complex structural features by combining these hyperedges.

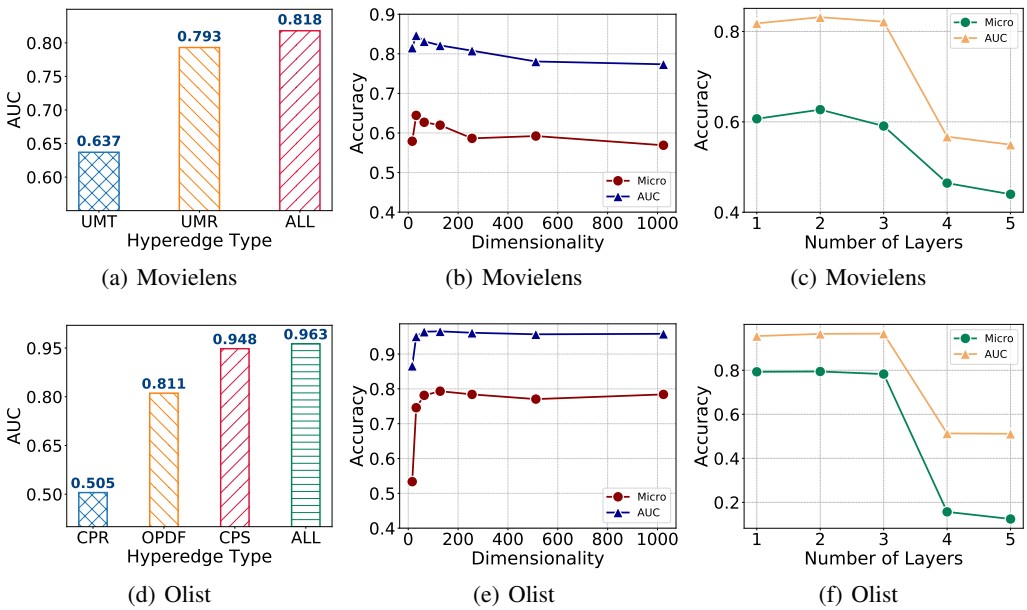

Figure 6: Hyper-parameter study on Movielens and Olist.

## 5.5 Hyper-parameter Study

Our proposed HHNN does not have special hyper-parameters. Here, we study the sensitivity of the hidden dimensionality and the number of model layers on the two datasets, and the results are shown in Figure 6.

Referring to Figure 6(b) and Figure 6(e), the model performance is poor when the hidden dimensionality is too small, which indicates that the model suffers from underfitting. HHNN shows the best results when the dimensionalities are equal to 64 on Movielens and 128 on Olist, respectively. After that, the model's performance gradually decreases as the dimensionality increases, which is caused by the overfitting issue.

Referring to Figure 6(c) and Figure 6(f), HHNN achieves its best performance when it has two model layers, and the model maintains strong performance even at 3 layers, with no significant performance drop, indicating robustness against over-smoothing. After that, the model's performance decreases significantly when the model goes deeper, which is caused by the overfitting issue of the hypergraph neural network. Notably, each HHNN layer includes two projection and aggregation steps (node-to-hyperedge and back), which actually correspond to two layers in standard GNN models. Therefore, a 3-layer HHNN is equivalent to a 6-layer standard GNN, making it sufficiently deep to capture complex structural patterns without degradation.

## 6 Conclusion

In this work, we first define a novel concept called hyper-meta-path for heterogeneous hypergraphs. It not only describes more complex structural features but also conveys richer semantic information. Then, we design a three-level attention-based heterogeneous hypergraph neural network called HHNN to automatically learn the importance of hyper-meta-paths. By discovering and exploiting useful ones, HHNN can achieve higher performance, and the semantics conveyed by these hyper-meta-paths can enhance the model interpretability of HHNN.

## Acknowledgments

This work was supported in part by the National Natural Science Foundation of China under Grants 62303366, 62425605, and 62133012, in part by the Key Research and Development Program of Shaanxi under Grants 2025CY-YBXM-041, 2022ZDLGY01-10, and 2024CY2-GJHX-15, and in part by the Fundamental Research Funds for the Central Universities under Grants ZYTS25211 and ZYTS25086.

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

# A    Datasets

Most previous studies [12, 7, 2, 6, 18, 21, 14, 38, 30, 3] construct their hypergraphs based on Cora, PubMed, and Citeseer, which originally belong to homogeneous simple graphs (as shown in Figure 1(a)). Several methods [5, 10, 22] also construct their hypergraphs based on DBLP, ACM, MAG, and IMDB, which originally belong to heterogeneous simple graphs (as shown in Figure 1(b)). *In other words, these hypergraphs are transformed from simple graphs, and they essentially describe binary relations rather than multi-ary relations.*

In this work, to fully reflect the effectiveness of our proposed HHNN in capturing complex structural features conveyed by hyper-meta-paths, we have tried our best to construct two genuine heterogeneous hypergraphs, where there are not only multiple types of nodes but also multiple types of real hyper-relations, as illustrated by Figure 1(d). To this end, we have tried our best to construct two new heterogeneous hypergraph benchmark datasets, and their key statistics are listed in Table 4.

Table 4: Dataset statistics.

| Datasets | Node Type | Number | Hyperedge Type | Hyperedge Semantics | Number |
|---|---|---|---|---|---|
| Movielens | Movies (M) | 3439 | UMT ($R_1$) | "a user tags a movie with a specific tag" | 9681 |
| | User (U) | 2106 | | | |
| | Tag (T) | 3108 | UMR ($R_2$) | "a user rates a movie" | 158897 |
| | Rating (R) | 10 | | | |
| Olist | Customer (C) | 54321 | CPS ($R_1$) | "a customer purchases a product from a seller" | 55630 |
| | Product (P) | 18454 | | | |
| | Seller (S) | 2090 | CPR ($R_2$) | "a customer provides a rating for a product" | 55556 |
| | Rating (R) | 5 | | | |
| | Origin (O) | 489 | OPDF ($R_3$) | "an order is shipped from an origin to a destination with a freight value" | 55630 |
| | Destination (D) | 3477 | | | |
| | Freight (F) | 14 | | | |

•**MovieLens** is a real-world movie dataset, which was originally released by GroupLens[2], a research laboratory at University of Minnesota. Based on the raw dataset, we construct a heterogeneous hypergraph that contains four types of nodes and two types of hyperedges. As we can see, the two hyperedge types (hyper-relations $R_1$ and $R_2$) are natural ternary relations. Movie nodes are associated with ground-truth labels, describing their genres, such as comedy, action, animation, etc.

•**Olist** is a real-world e-commercial dataset. It contains the orders of Olist Store, a Brazilian e-commerce platform. The raw dataset was originally released at Kaggle[3], a data science competition platform. Based on the raw dataset, we construct a heterogeneous hypergraph that contains six types of nodes and three types of hyperedges. As shown in the table, the hyper-relations $R_1$ and $R_1$ are ternary relations, and the hyper-relation $R_1$ is a quaternary relation. The product nodes are associated with ground-truth labels that describe their categories, such as perfumery, telephone, automotive, etc.

# B    Baselines

We select ten representative hypergraph neural network methods as our comparative baselines, including: (1) CE-GCN; (2) HNHN [7]; (3) HyperGCN [38]; (4) AllSetTsfm [6]; (5) AllDeepSets [6]; (6) UniGNN [21]; (7) ED-HNN [34]; (8) HGNN [12]; (9) HGNN+ [14].

Here, CE-GCN is a naive baseline, which first transforms a hypergraph into a simple graph through clique expansion [1], where two nodes form an edge if and only if they are in the same hyperedge. Then, the GCN [24] model is utilized to encode the resulting simple graph. For baseline UniGNN, we adopt its most powerful variant, i.e., UniGCNII. Baselines AllSetTsfm (AllSetTransformer) and AllDeepSets are two variants of the AllSet method [6].

---

[2]`https://files.grouplens.org/datasets/hetrec2011/hetrec2011-movielens-2k-v2.zip`
[3]`https://www.kaggle.com/datasets/olistbr/brazilian-ecommerce`

## C  Experimental Settings

We use the Pytorch framework to implement our proposed HHNN. The number of model layers is searched in $\{1, 2, 3, 4, 5\}$, the dimensionality of hidden node/hyperedge representations is searched in $\{8, 16, 32, 64, 128\}$, the number of attention heads is searched in $\{1, 2, 4, 8\}$. We use the Adam optimizer to optimize all the trainable model parameters, which are randomly initialized by the Xavier uniform distribution [16]. For ease of tuning, the optimizer settings are the same for both datasets. Specifically, the learning rate is set to 0.001, the weight decay is set to 0.0, and the attention dropout rate is set to 0.5. For more details about hyper-parameter settings, please see our source code at: `https://github.com/zhengziyu77/HHNN`.

Regarding baselines, we reproduce the experimental results on the two benchmark datasets, based on their official source codes. For fairness, we also try our best to search for the optimal hyper-parameters for them, starting from the default settings of their code or the settings reported in their papers.

For fairness, some of the settings are shared among all the methods. Firstly, they use the same input features. Secondly, they use the same evaluation metrics of Micro F1 (Micro) score, Macro F1 (Macro) score, and Area Under the Curve (AUC) score. Finally, they use the same training/validation/test sets. Specifically, on each dataset, we randomly select $\tau\%$ ground-truth labels as the training set, and the rest $(1 - \tau)\%$ are divided equally as the validation set and the test set, where $\tau \in \{20, 40\}$. We randomly repeat all the evaluation tasks ten times and report the mean results with the standard deviation. All the experiments are conducted on Intel(R) Core(TM) i9-10980XE CPU and NVIDIA TITAN RTX GPU with 24GB GPU memory.

## D  Algorithm

The overall training process of our proposed HHNN is shown in Algorithm 1.

---

**Algorithm 1:** The training process of HHNN.

---

**Input** : The heterogeneous hypergraph $\mathcal{G} = (\mathcal{V}, \mathcal{E}, \mathbf{H}, \mathbf{W}, \phi, \psi)$,
   The number of model layers $N$.
**Output :** The embeddings of all the nodes and hyperedges.

---

1 Randomly initialize all the trainable model parameters;
2 **for** $n = 1, ..., N$ **do**
3 $\quad$ # from nodes to hyperedges;
4 $\quad$ Perform $\alpha$-Attention aggregation according to Eqs. (1-3);
5 $\quad$ # from hyperedges to nodes;
6 $\quad$ Perform $\beta$-Attention (intra-type) aggregation according to Eqs. (4-6);
7 $\quad$ Perform $\gamma$-Attention (inter-type) aggregation according to Eqs. (7-8);
8 **end**
9 Compute loss according to Eq. (9);
10 Update model parameters by gradient descent;

---

## E  Theoretical Analysis of Hyper-meta-paths

The HHNN model introduces a hierarchical attention mechanism at three levels, i.e., node-to-hyperedge, intra-type hyperedge-to-node, and inter-type hyperedge-to-node, which correspond to the $\alpha$-Attention, the $\beta$-Attention, and the $\gamma$-Attention, respectively. Here, we can theoretically show that this attention structure enables the model to automatically discover important hyper-meta-paths by back-tracing the learned attention coefficients.

### E.1  Hyper-meta-path Importance Calculation

Consider a target node $i \in \mathcal{V}$, its final embedding is output by the last layer $N$, which reflects the information that flows through a sequence of interconnected nodes and hyperedges across the $N$ layers. Theoretically, the importance is the product of the involved attention weights along this

sequence. Thus, regarding the hyper-meta-path $\mathcal{P} = R_1 \diamond R_2 \diamond \cdots \diamond R_N$ involved in this sequence, its importance can be computed as follows:

$$I(\mathcal{P}) = \prod_{l=1}^{N} \alpha^{[l]} \cdot \beta^{[l]} \cdot \gamma^{[l]} \tag{10}$$

A higher value of $I(\mathcal{P})$ indicates that the information that propagates along the hyper-meta-path $\mathcal{P}$ contributes more significantly to the target nodes $i$'s final embedding $v_i^{[N]}$.

### E.2 Important Hyper-meta-path Discovering

By calculating the importance $I(\mathcal{P})$ for all relevant paths that lead to the target node $i$, we can identify the most important hyper-meta-path $\mathcal{P}^*$ for node $i$ as follows:

$$\mathcal{P}^* = \arg\max_{\mathcal{P}} I(\mathcal{P}) \tag{11}$$

$\mathcal{P}^*$ represents the most significant hyper-meta-path for the target node $i$, as learned by the model.

### E.3 Interpretability of Discoverd Hyper-meta-path

The automatically discovered hyper-meta-path $\mathcal{P}^*$ carries specific semantic meaning derived from the composition of the involved individual hyper-relations. This semantic interpretation can be used to explain the model's prediction for the target node $i$. For instance, if the task is node classification, the semantics of $\mathcal{P}^*$ can provide insight into why the model assigned a particular class label to the target node $i$.

### E.4 Discussion and Outlook of Attention-based Hyper-meta-path Discovery

HHNN discovers hypermeta-paths based on the attention distribution, and this attention-based heuristic reveals correlations rather than causal relationships. In the future work, formalization of causal interpretability is an important direction, and we plan to explore techniques like counterfactual reasoning in our future work. Here, we also conduct a preliminary experiment of masking information along high-weight paths and observing the change in the model's prediction, and the results are shown in the Table 5. As we can observe, by masking information along high-weight paths, the model's performance drops sharply, which clearly indicates that the high-weight paths identified by the HHNN model are crucial.

Table 5: Comparison between HHNN performance with high-weight paths masked (right) and the original performance (left) under the 20% training ratio.

|  | Movielen | Olist |
|---|---|---|
| Micro | $56.80_{\pm 1.65} \rightarrow 28.63_{\pm 5.32}$ | $72.54_{\pm 0.54} \rightarrow 13.24_{\pm 0.81}$ |
| Macro | $26.24_{\pm 3.90} \rightarrow 11.83_{\pm 2.80}$ | $72.89_{\pm 0.56} \rightarrow 6.38_{\pm 0.73}$ |
| AUC | $73.21_{\pm 2.13} \rightarrow 54.04_{\pm 1.31}$ | $93.20_{\pm 0.32} \rightarrow 50.13_{\pm 0.51}$ |

## F Comparison between HAN and HHNN

We compare our HHNN to HAN, which is based on a two-level attention aggregation mechanism. The results are shown in Table 6. As we can see, our HHNN can achieve higher performance against HAN on both ACM and DBLP, which demonstrates the superiority of our HHNN.

## G Experiments on More Datasets

Most existing heterogeneous hypergraph datasets are transformed from simple graphs with binary relations, which lack natural multi-way interactions and cannot fully showcase the advantages of

Table 6: Dataset statistics.

|  | ACM | DBLP |
|---|---|---|
| HAN | $85.09_{\pm 2.48}$ | $90.02_{\pm 0.87}$ |
| **HHNN** | $\textbf{87.28}_{\pm 0.88}$ | $\textbf{91.29}_{\pm 0.92}$ |

hypergraph neural networks. In contrast, in our work, we have tried our best to construct two new real-world datasets from the movie domain and e-commerce domain, both containing multiple types of naturally occurring heterogeneous hyperedges based on genuine multi-ary relations. The two datasets are more challenging to fully showcase the power of our HHNN. Here, we also conduct a performance comparison on three widely-used hypergraph datasets, i.e., Cora, PuMmed, and DBLP, and the results are shown in Table 7. We can observe that since these three datasets do not contain complex and heterogeneous structural features, the three hypergraph methods achieve comparable performance, with our HHNN performing slightly better.

Table 7: The performance comparison between three hypergraph methods on three new hypergraph datasets, under the 20% training ratio.

|  | Cora | PubMed | DBLP |
|---|---|---|---|
| UniGNN | $76.93_{\pm 1.73}$ | $84.05_{\pm 0.63}$ | $90.49_{\pm 0.18}$ |
| ED-HNN | $77.07_{\pm 1.22}$ | $84.12_{\pm 0.68}$ | $90.50_{\pm 0.21}$ |
| **HHNN** | $\textbf{77.42}_{\pm 1.67}$ | $\textbf{84.14}_{\pm 0.74}$ | $\textbf{90.50}_{\pm 0.27}$ |

Table 8: The performance comparison on Movielens, under the 20% training ratio.

|  | Micro | Macro | AUC |
|---|---|---|---|
| SheafHyperGNN [9] | $46.59_{\pm 1.24}$ | $18.85_{\pm 1.74}$ | $61.68_{\pm 1.48}$ |
| UniG-Encoder [44] | $41.79_{\pm 1.17}$ | $16.15_{\pm 0.96}$ | $58.66_{\pm 1.32}$ |
| **HHNN** | $\textbf{56.80}_{\pm 1.65}$ | $\textbf{26.24}_{\pm 3.90}$ | $\textbf{73.21}_{\pm 2.13}$ |

Table 9: The performance comparison on Olist, under the 20% training ratio.

|  | Micro | Macro | AUC |
|---|---|---|---|
| SheafHyperGNN [9] | $51.71_{\pm 1.54}$ | $51.56_{\pm 1.38}$ | $85.60_{\pm 1.02}$ |
| UniG-Encoder [44] | $24.40_{\pm 0.39}$ | $23.68_{\pm 0.49}$ | $64.96_{\pm 0.28}$ |
| **HHNN** | $\textbf{72.54}_{\pm 0.54}$ | $\textbf{72.89}_{\pm 0.56}$ | $\textbf{93.20}_{\pm 0.32}$ |

Table 10: The running time (seconds of 10 epochs) of HHNN on heterogeneous hypergraphs of different sizes using varying ratios of nodes.

|  | Movielens | Olist |
|---|---|---|
| 20% | 0.730 | 2.361 |
| 40% | 0.763 | 2.449 |
| 60% | 1.114 | 2.839 |
| 80% | 1.205 | 2.883 |
| 100% | 1.231 | 3.194 |

## H  Experiments with More Baselines

We compare our HHNN against two recent hypergraph models, i.e., SheafHyperGNN [9] and UniG-Encoder [44], as the new baselines, and their experimental results are shown in Table 8 and Table 9.

Table 11: The running time (seconds of 10 epochs) of HHNN on heterogeneous hypergraphs of different sizes using varying ratios of hyperedges.

| | Movielens | Olist |
|---|---|---|
| 20% | 0.653 | 2.413 |
| 40% | 0.805 | 2.528 |
| 60% | 0.927 | 2.809 |
| 80% | 1.111 | 2.978 |
| 100% | 1.231 | 3.194 |

Table 12: The running time (seconds of 1 epoch) of HHNN in comparison with baselines UniGNN and EDHNN.

| | HyperGCN | UniGNN | ED-HNN | HHNN |
|---|---|---|---|---|
| Movielens | 81.57 | 0.13 | 0.18 | 0.15 |
| Olist | 90.22 | 0.17 | 0.63 | 0.19 |

We can see that our HHNN significantly outperforms the two hypergraph baseline methods on both datasets.

## I Efficiency Study

To further verify the efficiency of HHNN, we have added the scalability experiment as well as the running time comparison experiment. The results are shown in Table 10, Table 11, and Table 12. As we can see, the running time of HHNN increases approximately linearly with the size of the graph. Besides, the running time of HHNN is on par with two baselines, UniGNN and EDHNN, and is much less than the running time of the baseline HyperGCN.

## J More Related Work

In the past decade, the representation learning methods on hypergraphs have become a research surge. In the recent two years, several surveys [1, 15, 32, 23, 28] have been published to comprehensively review existing hypergraph representation learning methods from different perspectives.

Regarding the heterogeneity of nodes and hyperedges, some of the existing methods, e.g., [4], only consider node heterogeneity, and some methods generally consider the hyperedge heterogeneity based on operations like concatenation [30] and adaptive fusion [14]. RelBench [13] leverages graph neural networks to learn directly from relational databases, which can also be viewed as a heterogeneous hypergraph neural network. In contrast, HHNN focuses on hyper-meta-paths to capture more complex structural features.

There are also some other path-related methods. Pathsim [31] is the first work to define the concept of meta-path as a sequence of binary relations among two nodes, which has shown extraordinary effectiveness in capturing heterogeneous structural features and rich semantics contained in heterogeneous simple graphs. The subsequent study, metapath2vec [8], leverages a set of user-specified meta-paths to guide the random walk on heterogeneous simple graphs, transforming heterogeneous structural features into sequences. Then, it develops a word2vec [27]-like encoder to obtain the node representations. Considering that existing methods need users to specify useful meta-paths, which is not practical, ie-HGCN [40] is proposed to automatically discover and exploit useful meta-paths in heterogeneous simple graphs. HAN [35] is a well-known GNN method for heterogeneous simple graphs, which is based on a set of user-specified meta-paths. While our model and HAN both adopt hierarchical attention, they are fundamentally different in scope and purpose. Firstly, HAN operates on heterogeneous simple graphs, where all relations are binary between node pairs, whereas HHNN is designed for heterogeneous hypergraphs that naturally encode multi-ary relations of various types. Moreover, HAN's two-level attention relies on manually specified meta-paths, while HHNN introduces a three-level attention mechanism that automatically discovers hyper-meta-paths without

manual intervention. HMG-CR [39] proposes the concept of hyper meta-path, which is a composition of multiple meta-paths between two specified end nodes in a heterogeneous simple graph. HPHG [20] defines the concept of hyper-path, which is utilized to preserve the complex information about the indecomposability of hypergraphs.

Different from these existing methods above, in this work, we define a novel concept called hyper-meta-path, which describes a complex hyper-relation by compositing a sequence of hyper-relations.

## K  Limitations

While the proposed HHNN demonstrates strong performance in modeling heterogeneous hypergraphs and good ability to automatically discover semantically meaningful hyper-meta-paths, several limitations should be acknowledged: (1) Although the attention-based aggregation enhances interpretability, it reveals correlations rather than causal relationships; (2) Although the model can learn to ignore some meaningless combinations by the attention optimization, the current definition of hyper-meta-path may be two broad in real-world scenarios, particularly when the shared node type is generic or high-frequency; (3) HHNN is particularly designed for complex heterogeneous hypergraphs, and it is most pronounced on hypergraphs with natural multi-ary relations, and its advantages may not be fully realized when such structures are not prominent. In future work, we would like to further explore alternative definitions and discovery mechanisms for hyper-meta-paths, and the direction of utilizing causality is very promising.

## L  Broader Impacts

The proposed HHNN model contributes to the broader field of graph representation learning by enabling improved performance and understanding for heterogeneous hypergraphs. This advancement can benefit many applications, such as recommendation systems, bioinformatics, and social network analysis, by simultaneously improving task performance and enhancing model interpretability through the modeling of more complex and higher-order relational patterns.

