# OpenReview forum: "Defining and Discovering Hyper-meta-paths for Heterogeneous Hypergraphs"
_NeurIPS.cc/2025/Conference — NeurIPS 2025 poster_

### Official Review · Reviewer_qQ4P · 2025-06-30

**Clarity:** 3
**Significance:** 4
**Originality:** 4
**Rating:** 5
**Confidence:** 5

**Summary:**

This study presents a novel concept of hyper-meta-path for heterogeneous hypergraphs where there are multiple types of nodes as well as multiple types of hyperedges. To be more specific, the hyper-meta-path is defined as the composition of a sequence of hyper-relations, and each hyper-relation describes a multi-ary relation, corresponding to one hyperedge type. The hyper-meta-path is expected to capture more complex structural features of heterogeneous hypergraphs. The authors design an attention-based heterogeneous hypergraph neural network to learn the importance of different hyper-meta-paths, and conduct extensive experiments to verify the effectiveness of the proposed concept.

**Questions:**

My questions are the same as the four weaknesses listed above. Please address those weaknesses.

**Ethical Concerns:**

["NO or VERY MINOR ethics concerns only"]

**Final Justification:**

After checking the rebuttal, I believe my concerns have been well addressed, thus I vote for acceptance.

**Limitations:**

Yes

**Quality:**

4

**Strengths And Weaknesses:**

**Strengths**

- The proposed concept of hyper-meta-path is novel and interesting, and is well defined. The authors have carefully analyzed its effectiveness, and verified its effectiveness through extensive experiments.

- At the beginning of Section 1, the review of the existing four categories of graph data is well organized, and the four illustrations are intuitive.

- The proposed model HHNN is clearly described by equations and intuitively described by Figure 3 and Figure 4. Therefore, the technical description of the methodology is highly readable and easy to follow.

- The comparison experiments show that the performance of the proposed method is promising. The ablation study and hyper-meta-path discovery experiment show that the proposed methods can effectively discover useful hyper-meta-paths.

- The authors have provided the source code and the used datasets, and provided much detailed information in the appendix, facilitating good reproducibility of this work.



**Weaknesses**

- The time complexity of the proposed method could be analyzed.

- More baselines could be considered, and the related work could be further enriched.

- According to the content, it is more appropriate to update the subtitle of Section 4.3 to "Hyper-meta-path Discovery and Interpretability Study".

- It seems that there is a minor issue in Line 267. According to Figure 5(a), the {UMT} hyperedge and the {UMR} hyperedge are connected by a user node, instead of a movie node.

---

> ### Author Rebuttal · Authors · 2025-07-29
>
> Thanks for your comments. In the following, we respond to your concerns point by point.
>
> ---
>
> **Point 1**: *The time complexity of the proposed method could be analyzed.*
>
> **Reply 1**: Thanks for your kind suggestion. The time complexity of HHNN is analyzed as follows. For each layer, the main cost comes from the three-level attention aggregation operations. The $\alpha$-attention aggregation involves projecting and attending to nodes within each hyperedge, costing $\mathcal{O}(|E| \cdot d_\alpha \cdot f_\alpha \cdot r)$, where $|E|$ is the number of hyperedges, $d_\alpha$ and $f_\alpha$ are the dimensionalities of the input and output features, respectively, and $r$ denotes the average number of nodes per hyperedge. In the $\beta$-attention aggregation phase, each node aggregates its connected hyperedges of each type, costing $\mathcal{O}(|V| \cdot d_\beta \cdot f_\beta \cdot t)$ where $|V|$ is the number of nodes, $d_\beta$ and $f_\beta$ are the feature dimensionalities, and $t$ denotes the average number of hyperedges of each type that a node participates in. In the $\gamma$-attention aggregation phase, each node fuses its representations across different hyperedge types, costing $\mathcal{O}(|V| \cdot d_\gamma \cdot f_\gamma \cdot k)$, where $d_\gamma$ and $f_\gamma$ are the feature dimensionalities, and $k$ is the average number of hyperedge types. Assuming the model has $n$ layers, the total time complexity becomes: $\mathcal{O}(|E| \cdot d_\alpha \cdot f_\alpha \cdot r \cdot n + |V| \cdot d_\beta \cdot f_\beta \cdot t \cdot n + |V| \cdot d_\gamma \cdot f_\gamma \cdot k \cdot n)$. In practical scenarios, $d_\alpha, f_\alpha, d_\beta, f_\beta, d_\gamma, f_\gamma, r, t, k, n$ are usually much smaller constants compared to $|\mathcal{V}|$ and $|\mathcal{E}|$. Therefore, the total time complexity of HHNN is equal to: $\mathcal{O}(|\mathcal{V}| + |\mathcal{E}|)$, which is linear to the size of the hypergraph. In addition, to verify the efficiency of HHNN, we have added the scalability experiment as well as the running time comparison experiment. The results are shown in the three tables below.
>
> *Table (1). We induce heterogeneous hypergraphs of different sizes using *varying ratios of nodes*, and the running time (seconds of 10 epochs) of HHNN increases approximately linearly with the size of the graph.*
> |      | Movielens | Olist |
> | ---- | --------- | ----- |
> | 20%  | 0.730     | 2.361 |
> | 40%  | 0.763     | 2.449 |
> | 60%  | 1.114     | 2.839 |
> | 80%  | 1.205     | 2.883 |
> | 100% | 1.231     | 3.194 |
>
> *Table (2). We induce heterogeneous hypergraphs of different sizes using *varying ratios of hyperedges*, and the running time (seconds of 10 epochs) of HHNN increases approximately linearly with the size of the graph.*
> |      | Movielens | Olist |
> | ---- | --------- | ----- |
> | 20%  | 0.653     | 2.413 |
> | 40%  | 0.805     | 2.528 |
> | 60%  | 0.927     | 2.809 |
> | 80%  | 1.111     | 2.978 |
> | 100% | 1.231     | 3.194 |
>
> *Table (3). The running time (seconds of 1 epoch) of HHNN is on par with two baselines, UniGNN and EDHNN. Kindly remind that the efficiency was not claimed as our main contribution.*
> |           | UniGNN  | ED-HNN | HNHN |
> | --------- | ------- | ------ | ---- |
> | Movielens | 0.13    | 0.18   | 0.15 |
> | Olist     | 0.17    | 0.63   | 0.19 |
>
>
> ---
>
> **Point 2**: *More baselines could be considered, and the related work could be further enriched.*
>
> **Reply 2**: Thanks for your kind suggestion. We have added two recent hypergraph models, i.e., SheafHyperGNN [1] and UniG-Encoder [2], as the new baselines, and their experimental results are shown in the two tables below. We can see that our HHNN significantly outperforms the two hypergraph baselines on both datasets. We will also enrich our related work by introducing the two methods, as well as more recent heterogeneous hypergraph neural networks.
>
> *Table (4). The performance comparison on Movielens, under the 20% training ratio.*
> |                   | Micro          | Macro          | AUC            |
> | ----------------- | -------------- | -------------- | ----------     |
> | SheafHyperGNN [1] | 46.59±1.24     | 18.85±1.74     | 61.68±1.48     |
> | UniG-Encoder [2]  | 41.79±1.17     | 16.15±0.96     | 58.66±1.32     |
> | **HHNN**          | **56.80±1.65** | **26.24±3.90** | **73.21±2.13** |
>
> Table (5). The performance comparison on Olist, under the 20% training ratio.
> |                   | Micro          | Macro          | AUC            |
> | ----------------- | -------------- | -------------- | -------------- |
> | SheafHyperGNN [1] | 51.71±1.54     | 51.56±1.38     | 85.60±1.02     |
> | UniG-Encoder [2]  | 24.40±0.39     | 23.68±0.49     | 64.96±0.28     |
> | **HHNN**          | **72.54±0.54** | **72.89±0.56** | **93.20±0.32** |
>
>
> *Refs*:
>
> [1] Duta, Iulia, et al. "Sheaf hypergraph networks." Advances in Neural Information Processing Systems 36 (2023): 12087-12099.
>
> [2] Zou, Minhao, et al. "Unig-encoder: A universal feature encoder for graph and hypergraph node classification." Pattern Recognition 147 (2024): 110115.
>
> ---
>
> **Point 3**: *Update the subtitle of Section 4.3 to "Hyper-meta-path Discovery and Interpretability Study.*
>
> **Reply 3**: Thanks for your helpful suggestion. Section 4.3 conducts an interpretability study based on the useful hyper-meta-paths discovered by HHNN. Therefore, following your suggestion, we will update the subsection title to “Hyper-meta-path Discovery and Interpretability Study”.
>
> ---
>
> **Point 4**: *According to Figure 5(a), the {UMT} hyperedge and the {UMR} hyperedge are connected by a user node.*
>
> **Reply 4**: Thank you for your careful review. We are sorry for this typo. Indeed, the {UMT} hyperedge and the {UMR} hyperedge are connected by a shared user node. We will correct this typo in the new version of our manuscript.

---

> > ### Comment · Reviewer_qQ4P · 2025-08-06
> >
> > Thanks for the rebuttal. My concerns have been well addressed, thus I keep my original score and vote for acceptance.

---

> ### Author Response · Authors · 2025-08-09
>
> **Dear Reviewer qQ4P,**
>
> **We especially thank you for your constructive comments, which have helped us significantly improve the quality of our paper. We assure you that we will incorporate these discussions into the PDF version of the paper in the next revision. Finally, we are deeply grateful for your encouraging support of our work, which will motivate us to conduct more interesting research in the future.**

---

### Official Review · Reviewer_FZj8 · 2025-07-02

**Clarity:** 2
**Significance:** 2
**Originality:** 2
**Rating:** 4
**Confidence:** 2

**Summary:**

This paper addresses the important problem of capturing complex structural and semantic information in heterogeneous hypergraphs, which arises from the composition of different hyper-relations. To this end, the authors introduce a new concept, the "hyper-meta-path," as a generalization of the well-known "meta-path" from heterogeneous simple graphs. Based on this concept, the authors propose a novel model, HHNN (Heterogeneous Hypergraph Neural Network). The core of HHNN is a three-level attention mechanism (node-to-hyperedge, intra-type hyperedge-to-node, and inter-type hyperedge-to-node). This architecture is designed to automatically learn the importance of different hyper-meta-paths without requiring manual specification, thereby aiming to improve both model performance and interpretability. The authors conduct experiments on two self-constructed datasets, showing that HHNN outperforms several baseline methods.

**Questions:**

1. On the Scope and Rigor of Experimental Evaluation: I strongly advice to benchmark HHNN on widely used public datasets, even if they are derived from simple graphs (e.g., hypergraph versions of Cora, PubMed, DBLP, or a dataset from the TGB suite).

2. On Scalability and Computational Complexity: A three-level attention mechanism is intuitively expensive. Add an experiment that plots the training time (or throughput) as the graph size (nodes/hyperedges) increases. This is a standard and necessary experiment to understand the practical limits of the proposed method.

3. On the Definition of Hyper-meta-path: The current definition for composing hyper-relations (Rx ∩ Ry ≠ ∅) is very broad. It allows for composition as long as any node type is shared. Please discuss the potential limitations of this broad definition. For instance, if two hyper-relations only share a very generic and high-frequency node type (e.g., 'Timestamp' or 'Location'), their composition might be syntactically valid but semantically meaningless. Does your model implicitly learn to down-weight such compositions? A brief discussion of this point would add nuance to your work.

4. Over-smoothing: A standard HHNN layer involves two steps of aggregation (node-to-hyperedge and hyperedge-to-node). Deeper GNN models often suffer from over-smoothing, where node representations become indistinguishable. How does your model, especially when stacked for multiple layers, mitigate this well-known issue?

**Ethical Concerns:**

["NO or VERY MINOR ethics concerns only"]

**Final Justification:**

After reviewing the rebuttal and seeing the authors' carefully proposed revision plan, I believe the paper is a weak accept for the conference, at a confidence of 2.

**Limitations:**

Yes. The authors have adequately addressed the limitations and potential societal impacts of their work. They have dedicated specific sections in the appendix (Appendix G for Limitations and Appendix H for Broader Impacts) to discuss these aspects.

**Paper Formatting Concerns:**

I have reviewed the paper for formatting issues and have not found any major concerns.

**Quality:**

2

**Strengths And Weaknesses:**

Strengths：
1. Clear and Valuable Problem Definition: The paper accurately identifies a key limitation in existing representation learning methods for heterogeneous hypergraphs: they often focus on individual hyper-relations while ignoring the richer, composite semantics that emerge from their combinations. The motivation to extend the "meta-path" concept to this domain is well-founded and promising.
2.	Conceptual Contribution: The formal definition of the "hyper-meta-path" is a clear contribution of this work. It provides a new and useful theoretical lens for reasoning about and analyzing higher-order, heterogeneous dependencies within hypergraphs.

Weaknesses

1. Limited Methodological Novelty: The HHNN model's core architecture closely mirrors the HAN model's two-level attention mechanism (node-level and semantic-level). While HHNN adapts this for hypergraphs via β-Attention (intra-type) and γ-Attention (inter-type), it lacks fundamental innovation expected for top-tier venues like NeurIPS.

2. Weak Theoretical Grounding for Path Discovery: The claimed "path discovery" mechanism relies on attention weight products (Appendix C, Eq. 10), a heuristic approach common in GNNs but lacking rigorous theoretical justification. Without evidence linking high-weight paths to causal model behavior, the interpretability contribution is weakened.

3. Narrow Empirical Validation: Experiments are limited to two new datasets, raising concerns about potential bias. Benchmarking on standard hypergraph datasets (e.g., hypergraph Cora/PubMed) would better demonstrate generalizability and superiority over SOTA methods.

4. Lack of Computational Analysis: The three-level attention mechanism likely incurs high computational costs, yet the paper omits complexity analysis or runtime experiments. A brief mention of overhead (Appendix G) is insufficient, undermining the model's practical scalability.

---

> ### Author Rebuttal · Authors · 2025-07-29
>
> Thanks for your comments. In the following, we respond to your concerns point by point.
>
> ---
>
> **Point 1**: *HHNN model's core architecture closely mirrors the HAN model's two-level attention mechanism.*
>
> **Reply 1**: Thank you for the comment. While our model and HAN both adopt hierarchical attention, they are fundamentally different in scope and purpose. Firstly, HAN operates on heterogeneous simple graphs, where all relations are binary between node pairs, whereas HHNN is designed for heterogeneous hypergraphs that naturally encode multi-ary relations of various types. Moreover, HAN’s two-level attention relies on manually specified meta-paths, while HHNN introduces a three-level attention mechanism that automatically discovers hyper-meta-paths without manual intervention. Most importantly, our key contribution lies in defining a new structural pattern, i.e., hyper-meta-paths, on heterogeneous hypergraphs. As supported by our theoretical analysis and extensive experiments, this pattern not only effectively captures rich structural information in heterogeneous hypergraphs but also brings good interpretability for mode predictions. Our work offers a simple yet effective first step toward discovering these patterns, and we remain open and enthusiastic to more advanced methods for hyper-meta-path discovery in future research.
>
> ---
>
> **Point 2**: *Lack of theoretical justification, and lack of evidence linking high-weight paths to causal model behavior.*
>
> **Reply 2**: Thanks for your comment. We provided theoretical justification in Appendix C. Specifically, we formally define how the hierarchical attention mechanism ($\alpha$, $\beta$, $\gamma$) induces a distribution over hyper-meta-paths traversed during message passing. In addition, in Section 4.3, we show case studies where high-weight hyper-meta-paths align with human-understandable structural patterns. While this does not prove causal influence in a strict sense, it offers intuitive and evidence-backed interpretability. We acknowledge that further formalization of causal interpretability is an important direction, and we plan to explore techniques like counterfactual reasoning in our future work.
>
> ---
>
> **Point 3**: *Experiments are limited to two new datasets. Benchmark HHNN on standard hypergraph datasets, even if they are derived from simple graphs (e.g., hypergraph versions of Cora, PubMed, DBLP, or a dataset from the TGB suite).*
>
> **Reply 3**: Thank you for the comment. Most existing heterogeneous hypergraph datasets are transformed from simple graphs with binary relations, which lack natural multi-way interactions and cannot fully showcase the advantages of hypergraph neural networks. In contrast, we have tried our best to construct two new real-world datasets from the movie domain and e-commerce domain, both containing multiple types of naturally occurring heterogeneous hyperedges based on genuine multi-ary relations. Nevertheless, according to your helpful suggestion, we have added three new hypergraph datasets, i.e., Cora, PuMmed, and DBLP, and the results are shown in the table below.
>
> *Table I. The performance comparison between three hypergraph methods on three new hypergraph datasets, under the 20% training ratio. We can observe that since these three datasets do not contain complex and heterogeneous structural features, the three hypergraph methods achieve comparable performance, with our HHNN performing slightly better.*
> |            | Cora             | PubMed           | DBLP             |
> | ---------- | ---------------- | ---------------- | ---------------- |
> | UniGNN     | 76.93 ± 1.73     | 84.05 ± 0.63     | 90.49 ± 0.18     |
> | ED-HNN     | 77.07 ± 1.22     | 84.12 ± 0.68     | 90.50 ± 0.21     |
> | **HHNN**   | **77.42 ± 1.67** | **84.14 ± 0.74** | **90.50 ± 0.27** |
>
> ---
>
> **Point 4**: *Lack of complexity analysis, and lack of scalability experiment.*
>
> **Reply 4**: Thanks for your kind suggestion. The time complexity of HHNN is analyzed as follows. For each layer, the main cost comes from the three-level attention aggregation operations. The $\alpha$-attention aggregation involves projecting and attending to nodes within each hyperedge, costing $\mathcal{O}(|E| \cdot d_\alpha \cdot f_\alpha \cdot r)$, where $|E|$ is the number of hyperedges, $d_\alpha$ and $f_\alpha$ are the dimensionalities of the input and output features, respectively, and $r$ denotes the average number of nodes per hyperedge. In the $\beta$-attention aggregation phase, each node aggregates its connected hyperedges of each type, costing $\mathcal{O}(|V| \cdot d_\beta \cdot f_\beta \cdot t)$ where $|V|$ is the number of nodes, $d_\beta$ and $f_\beta$ are the feature dimensionalities, and $t$ denotes the average number of hyperedges of each type that a node participates in. In the $\gamma$-attention aggregation phase, each node fuses its representations across different hyperedge types, costing $\mathcal{O}(|V| \cdot d_\gamma \cdot f_\gamma \cdot k)$, where $d_\gamma$ and $f_\gamma$ are the feature dimensionalities, and $k$ is the average number of hyperedge types. Assuming the model has $n$ layers, the total time complexity becomes: $\mathcal{O}(|E| \cdot d_\alpha \cdot f_\alpha \cdot r \cdot n + |V| \cdot d_\beta \cdot f_\beta \cdot t \cdot n + |V| \cdot d_\gamma \cdot f_\gamma \cdot k \cdot n)$. In practical scenarios, $d_\alpha, f_\alpha, d_\beta, f_\beta, d_\gamma, f_\gamma, r, t, k, n$ are usually much smaller constants compared to $|\mathcal{V}|$ and $|\mathcal{E}|$. Therefore, the total time complexity of HHNN is equal to: $\mathcal{O}(|\mathcal{V}| + |\mathcal{E}|)$, which is linear to the size of the hypergraph. In addition, to verify the efficiency of HHNN, we have added the scalability experiment as well as the running time comparison experiment. The results are shown in the three tables below.
>
> *Table II. We induce heterogeneous hypergraphs of different sizes using *varying ratios of nodes*, and the running time (seconds of 10 epochs) of HHNN increases approximately linearly with the size of the graph.*
> |      | Movielens | Olist |
> | ---- | --------- | ----- |
> | 20%  | 0.730     | 2.361 |
> | 40%  | 0.763     | 2.449 |
> | 60%  | 1.114     | 2.839 |
> | 80%  | 1.205     | 2.883 |
> | 100% | 1.231     | 3.194 |
>
> *Table III. We induce heterogeneous hypergraphs of different sizes using *varying ratios of hyperedges*, and the running time (seconds of 10 epochs) of HHNN increases approximately linearly with the size of the graph.*
> |      | Movielens | Olist |
> | ---- | --------- | ----- |
> | 20%  | 0.653     | 2.413 |
> | 40%  | 0.805     | 2.528 |
> | 60%  | 0.927     | 2.809 |
> | 80%  | 1.111     | 2.978 |
> | 100% | 1.231     | 3.194 |
>
> *Table IV. The running time (seconds of 1 epoch) of HHNN is on par with two baselines, UniGNN and EDHNN, and is much less than the running time of the baseline HyperGCN. Kindly remind that the efficiency is not our main contribution.*
> |           | HyperGCN | UniGNN  | ED-HNN | HNHN |
> | --------- | -------- | ------- | ------ | ---- |
> | Movielens | 81.57    | 0.13    | 0.18   | 0.15 |
> | Olist     | 90.22    | 0.17    | 0.63   | 0.19 |
>
> ---
>
> **Point 5**: *The definition for composing hyper-relations is very broad. If two hyper-relations only share a very generic and high-frequency node type, their composition might be semantically meaningless.*
>
> **Reply 5**: Thank you for this thoughtful observation. We agree that the compositional condition is broad and may include semantically weak or spurious hyper-meta-paths, especially when the shared node type is generic. Nevertheless, HHNN can mitigate this issue through its hierarchical attention mechanism. At the intra-type level ($\beta$-attention), the model learns to weight different hyperedges based on their contextual importance to each node. At the inter-type level ($\gamma$-attention), it further aggregates across different hyperedge types, allowing the model to down-weight weak or generic compositions that are less informative for the task. Most importantly, the final influence of a hyper-meta-path is determined by the product of attention weights across all levels (see Eq. (10) in Appendix C), which implicitly suppresses semantically meaningless paths by assigning them a low cumulative weight. This design allows flexible path discovery, and the model learns to emphasize meaningful compositions and ignore noisy ones based on data-driven evidence. We will discuss this point further in the manuscript.
>
> ---
>
> **Point 6**: *How does HHNN mitigate the over-smoothing issue, especially when stacked for multiple layers?*
>
> **Reply 6**: Thank you for the comment. HHNN mitigates over-smoothing primarily through its type-aware and structured three-level attention aggregation mechanism, which separates node-to-hyperedge and hyperedge-to-node message passing, and further decomposes the latter into intra-type and inter-type attention aggregation. This design limits the indiscriminate mixing of heterogeneous features and helps preserve type-specific and high-order structural information throughout the network. In addition, as the model depth experiment shown in Figure 6(c) and Figure 6(f), the model maintains strong performance even at 3 layers, with no significant performance drop, indicating robustness against over-smoothing. Notably, each HHNN layer includes two projection and aggregation steps (node-to-hyperedge and back), which actually correspond to two layers in standard GNN models. Therefore, a 3-layer HHNN is equivalent to a 6-layer standard GNN, making it sufficiently deep to capture complex structural patterns without degradation.

---

> > ### Comment · Reviewer_FZj8 · 2025-08-04
> >
> > Thank you for your detailed response to my review. The additional experiments and analyses have clarified some aspects of the work, which I appreciate. After carefully reading your rebuttal, I would like to offer the following follow-up points, which I hope will help you further improve the quality of your paper.
> >
> > 1. On the Generalizability of Empirical Validation:
> > I am grateful for the supplementary experiments on the standard datasets: Cora, PubMed, and DBLP. However, as your own results indicate, HHNN's performance is only "slightly better" than the baseline models on these datasets. While positive, this result reinforces one of my core concerns: the advantages of HHNN's complexity and its three-level attention mechanism appear to be most pronounced on your custom-built datasets (Movielens, Olist), which feature natural multi-ary relations.
> > When applied to hypergraphs derived from simple graphs that lack this complex heterogeneous structure, the performance improvement is not significant. This leaves the question of the model's generalizability open. Therefore, I suggest that you more precisely position your contribution in the paper. Specifically, you could explicitly state that HHNN is particularly effective for complex scenarios that inherently contain rich, genuine multi-ary interactions, and discuss that its advantages may not be fully realized when such structures are not prominent.
> >
> > 2. On Methodological Novelty and Theoretical Depth:
> > I understand and accept that HHNN aims to solve a more general problem than HAN (hypergraphs vs. simple graphs) and that it achieves automatic path discovery instead of requiring manual specification, which is a valuable contribution.
> > However, I still maintain that the core "path discovery" mechanism—identifying important paths by back-tracing the product of attention weights—remains a heuristic approach from a theoretical standpoint. In your response, you mention that high-weight paths correspond to human-understandable patterns, providing "evidence-backed interpretability." This effectively demonstrates a correlation in the model's behavior but does not rigorously establish causality. That is, we know the model attends to a path, but we cannot be certain how removing that path would causally affect the final prediction.
> > To make the claims of "path discovery" and "interpretability" more compelling, I suggest you could add a deeper discussion in the paper (or at least in the appendix). For instance:
> >  - Briefly discuss the limitations of this attention-based heuristic.
> >  - If feasible, consider a simple perturbation analysis (e.g., removing or masking information along a high-weight path and observing the change in the model's prediction). This would provide stronger evidence for interpretability.
> >
> > 3. On the Definition of Hyper-meta-path:
> > Thank you for explaining that the hierarchical attention mechanism implicitly down-weights semantically meaningless compositions. This is a pragmatic solution and a common approach in deep learning to let the model learn how to handle noise from a broad definition. Nonetheless, I believe this somewhat sidesteps the more fundamental issue with the definition itself. While your model can "learn" to ignore meaningless combinations, that does not mean the definition is optimal. Therefore, I still believe that a brief discussion in the paper—acknowledging the broadness of the definition and explaining how the model mitigates this issue via its attention mechanism—would make your work more rigorous and comprehensive.
> >
> > Although some of  my concerns are resolved, I believe a deeper and more critical reflection on these points would elevate the quality of this paper to an even higher standard. I am looking forward to your reply.

---

> ### Author Response · Authors · 2025-08-04
>
> Thank you for your detailed follow-up comments, and we are happy to know that some of your previous concerns are resolved. In the following, we respond to your follow-up points.
>
> ---
>
> **Point 7**: *HHNN appears to be most pronounced on hypergraphs with natural multi-ary relations, and its advantages may not be fully realized when such structures are not prominent.*
>
> **Reply 7**: Thank you for the comment, and we fully agree with your careful observation. Since our HHNN is particularly designed for complex heterogeneous hypergraphs, and it is most pronounced on hypergraphs with natural multi-ary relations. According to your kind and helpful suggestion, we will revise our manuscript to explicitly state that HHNN is particularly effective for complex scenarios that inherently contain rich, genuine multi-ary interactions. We promise that we will definitely add the discussion that the advantages of HHNN may not be fully realized when the hypergraph structures are not very complex or heterogeneous.
>
> ---
>
> **Point 8**: *HHNN's use of three-level attention to identify hypermeta-paths demonstrates a correlation in the model's behavior but does not rigorously establish causality. Consider a simple perturbation analysis, e.g., removing or masking information along a high-weight path and observing the change in the model's prediction. Briefly discuss the limitations of this attention-based heuristic.*
>
> **Reply 8**: Thanks for your constructive and detailed comment. We agree that the way HHNN discovers hypermeta-paths belongs to correlation rather than causality, and we promise that we will definitely add the limitation discussion to clarify that our attention-based heuristic reveals correlations rather than causal relationships. According to your specific suggestion, we have added a new experiment of masking information along high-weight paths and observing the change in the model's prediction, and the results, as well as the analysis, are shown in the Table below.
>
> *Table V. Comparison between the performance with high-weight paths masked (right) and the original performance (left) under the 20% training ratio. We can observe that by masking information along high-weight paths, the model's performance drops sharply, which clearly indicates that the high-weight paths identified by the HHNN model are crucial.*
> |       | Movielen                              | Olist                                 |
> | ----- | ------------------------------------- | ------------------------------------- |
> | Micro | 56.80±1.65 $\rightarrow$ 28.63 ± 5.32 | 72.54±0.54 $\rightarrow$ 13.24 ± 0.81 |
> | Macro | 26.24±3.90 $\rightarrow$ 11.83 ± 2.80 | 72.89±0.56 $\rightarrow$ 6.38 ± 0.73  |
> | AUC   | 73.21±2.13 $\rightarrow$ 54.04 ± 1.31 | 93.20±0.32 $\rightarrow$ 50.13 ± 0.51 |
>
> ---
>
> **Point 9**: *Add a brief discussion in the paper, acknowledging the broadness of the definition and explaining how the model mitigates this issue via its attention mechanism.*
>
> **Reply 9**: Thank you for your thoughtful comment. We acknowledge that the current definition of hyper-meta-path is broad, and such generality may introduce noisy or trivial compositions, particularly when the shared node type is generic or high-frequency. According to your specific suggestion, we will revise the paper to add the limitation discussion about the broadness of the definition, and include the discussion of how HHNN mitigates this issue via its attention mechanism. This is the first work to propose the concept of hyper-meta-path, along with a simple yet effective method to automatically discover such patterns. In future work, we will further explore alternative definitions and discovery mechanisms for hyper-meta-paths. The direction of utilizing causality, as you kindly recommended, is very promising, and we greatly appreciate your insightful suggestion.

---

> > ### Comment · Reviewer_FZj8 · 2025-08-05
> >
> > I appreciate the authors’ efforts in addressing my concerns through their rebuttal, which has alleviated some of my concern. However, before finalizing my recommendation, I would like to see a more concrete revision plan outlining how the promised changes will be implemented in the paper. Specifically, it would be helpful if the authors could clarify: 1) Where (e.g., sections, line numbers) the new discussions or experimental results will be added and 2) What exact modifications will be made (e.g., added sentences or revised paragraphs). A clear commitment to these revisions would strengthen my confidence in the paper. Once these details are provided, I would like to raise my score.

---

> ### Author Response · Authors · 2025-08-06
> **Revision Plan (Part I)**
>
> We sincerely thank you for your encouraging and detailed suggestions. We hereby make the following solemn commitment: once we are allowed to revise the PDF of our manuscript, we will carefully modify our manuscript strictly in accordance with the reviewer’s comments, as outlined in the concrete revision plan below.
>
> ---
>
> **Revision 1 at Appendix A "Related Work", Paragraph "Path-related Methods", Line 464**: We will add the discussion about the connection between HAN and our HHNN as follows.
>
> While our model and HAN both adopt hierarchical attention, they are fundamentally different in scope and purpose. Firstly, HAN operates on heterogeneous simple graphs, where all relations are binary between node pairs, whereas HHNN is designed for heterogeneous hypergraphs that naturally encode multi-ary relations of various types. Moreover, HAN’s two-level attention relies on manually specified meta-paths, while HHNN introduces a three-level attention mechanism that automatically discovers hyper-meta-paths without manual intervention. We compared the performance of our HHNN against HAN in the Appendix.
>
> ---
>
> **Revision 2 at Appendix**: We will add a new section titled "Comparison between HAN and HHNN" to present the experiment of comparing HAN and our HHNN, and the content is as follows.
>
> We compare our HHNN to HAN, which is based on two-level attention aggregation mechanism. The results are shown in the table below. As we can see, our HHNN can achieve higher performance against HAN on both ACM and DBLP, which demonstrates the superiority of our HHNN.
>
> *Table. The performance compaison our HAN and HHNN on ACM and DBLP.*
> |          | ACM            | DBLP           |
> | -------- | -------------- | -------------- |
> | HAN      | 85.09±2.48     | 90.02±0.87     |
> | **HHNN** | **87.28±0.88** | **91.29±0.92** |
>
> ---
>
> **Revision 3 at Appendix C "Theoretical Analysis of Hyper-meta-paths"**: We will add a new subsection titled "Discussion and Outlook of Attention-based Hyper-meta-path Discovery", and the content is as follows:
>
> Our HHNN discovers hypermeta-paths based on the attention distribution, and this attention-based heuristic reveals correlations rather than causal relationships. In the future work, formalization of causal interpretability is an important direction, and we plan to explore techniques like counterfactual reasoning in our future work. Here, we also conduct a preliminary experiment of masking information along high-weight paths and observing the change in the model's prediction, and the results are shown in the Table below. As we can observe, by masking information along high-weight paths, the model's performance drops sharply, which clearly indicates that the high-weight paths identified by the HHNN model are crucial.
>
> *Table V. Comparison between the performance with high-weight paths masked (right) and the original performance (left) under the 20% training ratio.*
> |       | Movielen                              | Olist                                 |
> | ----- | ------------------------------------- | ------------------------------------- |
> | Micro | 56.80±1.65 $\rightarrow$ 28.63 ± 5.32 | 72.54±0.54 $\rightarrow$ 13.24 ± 0.81 |
> | Macro | 26.24±3.90 $\rightarrow$ 11.83 ± 2.80 | 72.89±0.56 $\rightarrow$ 6.38 ± 0.73  |
> | AUC   | 73.21±2.13 $\rightarrow$ 54.04 ± 1.31 | 93.20±0.32 $\rightarrow$ 50.13 ± 0.51 |

---

> ### Author Response · Authors · 2025-08-06
> **Revision Plan (Part II)**
>
> **Revision 4 at Appendix**: We will add a new section titled "Experiments on More Datasets" to explain the current dataset and present the new experiment on three more datasets, and the content is as follows.
>
> Most existing heterogeneous hypergraph datasets are transformed from simple graphs with binary relations, which lack natural multi-way interactions and cannot fully showcase the advantages of hypergraph neural networks. In contrast, in our work, we have tried our best to construct two new real-world datasets from the movie domain and e-commerce domain, both containing multiple types of naturally occurring heterogeneous hyperedges based on genuine multi-ary relations. The two datasets are more challenging to fully showcase the power of our HHNN. Here, we also conduct a performance comparison on three widely-used hypergraph datasets, i.e., Cora, PuMmed, and DBLP, and the results are shown in the table below. We can observe that since these three datasets do not contain complex and heterogeneous structural features, the three hypergraph methods achieve comparable performance, with our HHNN performing slightly better.
>
> *Table. The performance comparison between three hypergraph methods on three new hypergraph datasets, under the 20% training ratio.*
> |            | Cora             | PubMed           | DBLP             |
> | ---------- | ---------------- | ---------------- | ---------------- |
> | UniGNN     | 76.93 ± 1.73     | 84.05 ± 0.63     | 90.49 ± 0.18     |
> | ED-HNN     | 77.07 ± 1.22     | 84.12 ± 0.68     | 90.50 ± 0.21     |
> | **HHNN**   | **77.42 ± 1.67** | **84.14 ± 0.74** | **90.50 ± 0.27** |
>
> ---
>
> **Revision 5 at Section 3, Line 204**: We will add a new subsection titled "Time Complexity Analysis", and the content is as follows.
>
> The time complexity of HHNN is analyzed as follows. For each layer, the main cost comes from the three-level attention aggregation operations. The $\alpha$-attention aggregation involves projecting and attending to nodes within each hyperedge, costing $\mathcal{O}(|E| \cdot d_\alpha \cdot f_\alpha \cdot r)$, where $|E|$ is the number of hyperedges, $d_\alpha$ and $f_\alpha$ are the dimensionalities of the input and output features, respectively, and $r$ denotes the average number of nodes per hyperedge. In the $\beta$-attention aggregation phase, each node aggregates its connected hyperedges of each type, costing $\mathcal{O}(|V| \cdot d_\beta \cdot f_\beta \cdot t)$ where $|V|$ is the number of nodes, $d_\beta$ and $f_\beta$ are the feature dimensionalities, and $t$ denotes the average number of hyperedges of each type that a node participates in. In the $\gamma$-attention aggregation phase, each node fuses its representations across different hyperedge types, costing $\mathcal{O}(|V| \cdot d_\gamma \cdot f_\gamma \cdot k)$, where $d_\gamma$ and $f_\gamma$ are the feature dimensionalities, and $k$ is the average number of hyperedge types. Assuming the model has $n$ layers, the total time complexity becomes: $\mathcal{O}(|E| \cdot d_\alpha \cdot f_\alpha \cdot r \cdot n + |V| \cdot d_\beta \cdot f_\beta \cdot t \cdot n + |V| \cdot d_\gamma \cdot f_\gamma \cdot k \cdot n)$. In practical scenarios, $d_\alpha, f_\alpha, d_\beta, f_\beta, d_\gamma, f_\gamma, r, t, k, n$ are usually much smaller constants compared to $|\mathcal{V}|$ and $|\mathcal{E}|$. Therefore, the total time complexity of HHNN is equal to: $\mathcal{O}(|\mathcal{V}| + |\mathcal{E}|)$, which is linear to the size of the hypergraph.

---

> ### Author Response · Authors · 2025-08-06
> **Revision Plan (Part III)**
>
> **Revision 6 at Appendix**: We will add a new section to present the efficiency experiment, and the content is as follows.
>
> To further verify the efficiency of HHNN, we have added the scalability experiment as well as the running time comparison experiment. The results are shown in the three tables below. As we can see, the running time of HHNN increases approximately linearly with the size of the graph. Besides, the running time of HHNN is on par with two baselines, UniGNN and EDHNN, and is much less than the running time of the baseline HyperGCN.*
>
> *Table. The running time (seconds of 10 epochs) of HHNN on heterogeneous hypergraphs of different sizes using *varying ratios of nodes.*
> |      | Movielens | Olist |
> | ---- | --------- | ----- |
> | 20%  | 0.730     | 2.361 |
> | 40%  | 0.763     | 2.449 |
> | 60%  | 1.114     | 2.839 |
> | 80%  | 1.205     | 2.883 |
> | 100% | 1.231     | 3.194 |
>
> *Table. The running time (seconds of 10 epochs) of HHNN on heterogeneous hypergraphs of different sizes using *varying ratios of hyperedges.*
> |      | Movielens | Olist |
> | ---- | --------- | ----- |
> | 20%  | 0.653     | 2.413 |
> | 40%  | 0.805     | 2.528 |
> | 60%  | 0.927     | 2.809 |
> | 80%  | 1.111     | 2.978 |
> | 100% | 1.231     | 3.194 |
>
> *Table. The running time (seconds of 1 epoch) of HHNN in comparison with two baselines, UniGNN and EDHNN.*
> |           | HyperGCN | UniGNN  | ED-HNN | HNHN |
> | --------- | -------- | ------- | ------ | ---- |
> | Movielens | 81.57    | 0.13    | 0.18   | 0.15 |
> | Olist     | 90.22    | 0.17    | 0.63   | 0.19 |
>
> ---
>
> **Revision 7 at Appendix G "Limitations"**: We will revise this section as follows.
>
> While the proposed HHNN demonstrates strong performance in modeling heterogeneous hypergraphs and good ability to automatically discover semantically meaningful hyper-meta-paths, several limitations should be acknowledged: (1) Although the attention-based aggregation enhances interpretability, it reveals correlations rather than causal relationships; (2) Although the model can learn to ignore some meaningless combinations by the attention optimization, the current definition of hyper-meta-path may be two broad in real-world scenarios, particularly when the shared node type is generic or high-frequency; (3) HHNN is particularly designed for complex heterogeneous hypergraphs, and it is most pronounced on hypergraphs with natural multi-ary relations, and its advantages may not be fully realized when such structures are not prominent. In future work, we would like to further explore alternative definitions and discovery mechanisms for hyper-meta-paths, and the direction of utilizing causality is very promising.
>
> ---
>
> **Revision 8 at Section 4.5 "Hyper-parameter Study", Lines 295-299**: We will revise this paragraph to discuss the over-smoothing issue, and the revised content is as follows.
>
> Referring to Figure 6(c) and Figure 6(f), HHNN achieves its best performance when it has two model layers, and the model maintains strong performance even at 3 layers, with no significant performance drop, indicating robustness against over-smoothing. After that, the model's performance decreases significantly when the model goes deeper, which is caused by the overfitting issue of the hypergraph neural network. Notably, each HHNN layer includes two projection and aggregation steps (node-to-hyperedge and back), which actually correspond to two layers in standard GNN models. Therefore, a 3-layer HHNN is equivalent to a 6-layer standard GNN, making it sufficiently deep to capture complex structural patterns without degradation.
>
> ---
>
> **Revision 9 for the whole manuscript**:
>
> We will make adjustments to other parts of the paper to ensure compliance with the length requirements, such as resizing figures and modifying the length or placement of other non-essential content, etc.

---

> ### Author Response · Authors · 2025-08-07
>
> Dear Reviewer FZj8,
>
> We have carefully followed your suggestions to formulate a detailed revision plan and have made a solemn commitment to implement these revisions. May we kindly ask if you could consider raising your score, as you mentioned in the previous comment? We are truly grateful for your valuable feedback, which has greatly helped us improve the quality of our paper.

---

> > ### Comment · Reviewer_FZj8 · 2025-08-08
> >
> > I appreciate your engagement. I will raise my score to 4 to support acceptance.

---

> ### Author Response · Authors · 2025-08-09
>
> **Dear Reviewer FZj8,**
>
> **We sincerely thank you for devoting so much of your valuable time and effort to engaging in thorough discussions with us to improve the quality of our paper. We truly appreciate that you have raised your score to 4 and supported the acceptance of our work.**

---

### Official Review · Reviewer_bvBy · 2025-07-03

**Clarity:** 3
**Significance:** 2
**Originality:** 4
**Rating:** 3
**Confidence:** 4

**Summary:**

This paper proposes the concept of hyper-meta-paths, a generalization of meta-paths to heterogeneous hypergraphs, and introduces HHNN, a GNN architecture designed to leverage hyper-meta-path semantics via multi-level attention. The authors claim that this framework captures richer relational structures and improves both performance and interpretability. Experimental results demonstrate improved performance over existing baselines across several datasets.

**Questions:**

Q1. Can you compare HHNN with any path-based model to clarify the benefit of hyper-meta-paths?
Q2. How do you deal with the rapid growth of hyper-meta-paths?
Q3. Is the full three-level attention necessary? Any ablation results?
Q4. Can you provide an example or visualization to support the interpretability claim?
Q5. How does HHNN generalize to more complex heterogeneous hypergraphs with increased hyperedge arity and semantic diversity?

**Ethical Concerns:**

["NO or VERY MINOR ethics concerns only"]

**Final Justification:**

My rating remains as in the current review, with no additional comments.

**Limitations:**

The proposed method presents scalability challenges due to the rapid growth of hyper-meta-paths in heterogeneous hypergraphs. The claim of interpretability is not supported by concrete examples or analysis. Furthermore, the evaluation is conducted on relatively simple datasets, limiting insight into the broader applicability of the proposed approach.

**Paper Formatting Concerns:**

None.

**Quality:**

2

**Strengths And Weaknesses:**

Strengths
S1. The paper introduces the notion of hyper-meta-paths; a generalization of meta-paths for heterogeneous hypergraphs. This is appropriate for modeling multi-relational hypergraph semantics.
S2. The proposed HHNN model uses a three-level attention aggregation scheme (node -> hyperedge -> node) to capture hyper-relational dependencies. The architecture is intuitive and well-structured.
S3. HHNN outputs interpretable attention weights over hyper-meta-paths, which provides insight into which relational structures are most influential during prediction; an underexplored property in hypergraph learning.

Weaknesses
W1. The paper does not provide empirical evidence that hyper-meta-paths are essential. There is no comparison with meta-path-based or path-based baseline hypergraph models, making it unclear whether their modeling improves performance.
W2. The number of hyper-meta-paths can grow rapidly, but the paper does not discuss how to limit or handle this. This may affect scalability on more complex graphs.
W3. It is unclear whether all components of the model are necessary. The paper does not test what happens if parts of the attention mechanism are removed or modified.
W4. The datasets used are relatively simple and may not be sufficient to demonstrate the benefit of modeling complex hyper-relational structures.
W5. The model claims to offer interpretability, but no example or case study is provided to show that the learned hyper-meta-paths are meaningful.

---

> ### Author Rebuttal · Authors · 2025-07-29
>
> Thanks for your comments. In the following, we respond to your concerns point by point.
>
> ---
>
> **Point 1**: *Provide empirical evidence that hyper-meta-paths are essential. Comparison with meta-path-based baseline models.*
>
> **Reply 1**: We would like to kindly remind the reviewer that we did provide empirical evidence from both the ablation study in Section 4.2 and the case study in Section 4.3. Firstly, as shown in Table 3 of Section 4.2, we compare eight variants of HHNN by selectively disabling the α, β, and γ attention layers, which are responsible for capturing and composing hyper-meta-paths. The full version (variant "111") consistently outperforms all others, showing the effectiveness of hyper-meta-path in promoting performance. Secondly, as shown in Figure 5 of Section 4.3, we analyze learned attention weights to identify interpretable hyper-meta-paths contributing to predictions. For example, on Movielens, HHNN discovers that the composed hyper-meta-path T ← {UMT} ⋄ {UMR} → M captures semantics: “*a tag $t$ is linked to a movie $m$ through a user $u$, and the user $u$ associates the tag $t$ to a movie, and gives the movie $m$ a rating*”. This semantics is helpful for interpreting the model's prediction results on the movie node $m$. Prior meta-path-based models (e.g., HAN) are designed for heterogeneous simple graphs and cannot directly handle hypergraphs' multi-ary relations. Nevertheless, according to your kind suggestion, we have compared our HHNN with the meta-path-based baseline, HAN, and the results are shown in the table below.
>
> *Table A. Our HHNN can achieve superior performance against HAN on both ACM and DBLP.*
> |          | ACM            | DBLP           |
> | -------- | -------------- | -------------- |
> | HAN      | 85.09±2.48     | 90.02±0.87     |
> | **HHNN** | **87.28±0.88** | **91.29±0.92** |
>
> ---
>
> **Point 2**: *The number of hyper-meta-paths can grow rapidly, which may affect scalability on more complex graphs. How does HHNN generalize to more complex heterogeneous hypergraphs with increased hyperedge arity and semantic diversity?**
>
> **Reply 2**: Thanks for your constructive suggestion. Let us first analyze the theoretical time complexity of HHNN. For each layer, the main cost comes from the three-level attention aggregation operations. The $\alpha$-attention aggregation involves projecting and attending to nodes within each hyperedge, costing $\mathcal{O}(|E| \cdot d_\alpha \cdot f_\alpha \cdot r)$, where $|E|$ is the number of hyperedges, $d_\alpha$ and $f_\alpha$ are the dimensionalities of the input and output features, respectively, and $r$ denotes the average number of nodes per hyperedge. In the $\beta$-attention aggregation phase, each node aggregates its connected hyperedges of each type, costing $\mathcal{O}(|V| \cdot d_\beta \cdot f_\beta \cdot t)$ where $|V|$ is the number of nodes, $d_\beta$ and $f_\beta$ are the feature dimensionalities, and $t$ denotes the average number of hyperedges of each type that a node participates in. In the $\gamma$-attention aggregation phase, each node fuses its representations across different hyperedge types, costing $\mathcal{O}(|V| \cdot d_\gamma \cdot f_\gamma \cdot k)$, where $d_\gamma$ and $f_\gamma$ are the feature dimensionalities, and $k$ is the average number of hyperedge types. Assuming the model has $n$ layers, the total time complexity becomes: $\mathcal{O}(|E| \cdot d_\alpha \cdot f_\alpha \cdot r \cdot n + |V| \cdot d_\beta \cdot f_\beta \cdot t \cdot n + |V| \cdot d_\gamma \cdot f_\gamma \cdot k \cdot n)$. In practical scenarios, $d_\alpha, f_\alpha, d_\beta, f_\beta, d_\gamma, f_\gamma, r, t, k, n$ are usually much smaller constants compared to $|\mathcal{V}|$ and $|\mathcal{E}|$. Therefore, the total time complexity of HHNN is equal to: $\mathcal{O}(|\mathcal{V}| + |\mathcal{E}|)$, which is linear to the size of the hypergraph. Therefore, HHNN can well scale to large heterogeneous hypergraphs with more complex semantics. In addition, to verify the efficiency of HHNN, we have added the scalability experiment as well as the running time comparison experiment. The results are shown in the three tables below.
>
> *Table B. We induce heterogeneous hypergraphs of different sizes using *varying ratios of nodes*, and the running time (seconds of 10 epochs) of HHNN increases approximately linearly with the size of the graph.*
> |      | Movielens | Olist |
> | ---- | --------- | ----- |
> | 20%  | 0.730     | 2.361 |
> | 40%  | 0.763     | 2.449 |
> | 60%  | 1.114     | 2.839 |
> | 80%  | 1.205     | 2.883 |
> | 100% | 1.231     | 3.194 |
>
> *Table C. We induce heterogeneous hypergraphs of different sizes using *varying ratios of hyperedges*, and the running time (seconds of 10 epochs) of HHNN increases approximately linearly with the size of the graph.*
> |      | Movielens | Olist |
> | ---- | --------- | ----- |
> | 20%  | 0.653     | 2.413 |
> | 40%  | 0.805     | 2.528 |
> | 60%  | 0.927     | 2.809 |
> | 80%  | 1.111     | 2.978 |
> | 100% | 1.231     | 3.194 |
>
> *Table D. The running time (seconds of 1 epoch) of HHNN is on par with two baselines, UniGNN and EDHNN, and is much less than the running time of the baseline HyperGCN. Kindly remind that the efficiency is not our main contribution.*
> |           | HyperGCN | UniGNN  | ED-HNN | HNHN |
> | --------- | -------- | ------- | ------ | ---- |
> | Movielens | 81.57    | 0.13    | 0.18   | 0.15 |
> | Olist     | 90.22    | 0.17    | 0.63   | 0.19 |
>
>
> ---
>
> **Point 3**: *Whether all the components of the model are necessary? Is the full three-level attention necessary? Any ablation results?*
>
> **Reply 3**: We are sorry for leading you to this misunderstanding. We did provide the ablation study in Section 4.2 to demonstrate the necessity of each module. Please see the experimental results in Table 3, the variant "111", which activates all three attention blocks, consistently outperforms all other variants, fully demonstrating that the full three-level attention is very necessary.
>
> ---
>
> **Point 4**: *The datasets are relatively simple.*
>
> **Reply 4**: Thank you for the comment. Most existing heterogeneous hypergraph datasets are transformed from simple graphs with binary relations, which lack natural multi-way interactions and cannot fully showcase the advantages of hypergraph neural networks. In contrast, we have tried our best to construct two new real-world datasets from the movie domain and e-commerce domain, both containing multiple types of naturally occurring heterogeneous hyperedges based on genuine multi-ary relations. According to your suggestion, we will continue seeking more such datasets, and we would greatly appreciate any specific recommendations about such datasets. Nevertheless, we have added three new hypergraph datasets, i.e., Cora, PuMmed, and DBLP, and the results are shown in the table below.
>
> *Table E. The performance comparison between three hypergraph methods on three new hypergraph datasets, under the 20% training ratio. We can observe that since these three datasets do not contain complex and heterogeneous structural features, the three hypergraph methods achieve comparable performance, with our HHNN performing slightly better.*
> |            | Cora             | PubMed           | DBLP             |
> | ---------- | ---------------- | ---------------- | ---------------- |
> | UniGNN     | 76.93 ± 1.73     | 84.05 ± 0.63     | 90.49 ± 0.18     |
> | ED-HNN     | 77.07 ± 1.22     | 84.12 ± 0.68     | 90.50 ± 0.21     |
> | **HHNN**   | **77.42 ± 1.67** | **84.14 ± 0.74** | **90.50 ± 0.27** |
>
>
> ---
>
> **Point 5**: *No example or case study is provided to show that the learned hyper-meta-paths are meaningful. Provide an example or visualization to support the interpretability claim.*
>
> **Reply 5**: Thank you for the comment. We would like to kindly remind the reviewer that we did provide two case studies in Section 4.3. Please see the visualization examples in Figure 5(a) and Figure 5(b), where we intuitively show the discovered useful hyper-meta-paths on Movielens and Olist, respectively. For example, on Movielens, HHNN discovers that the composed hyper-meta-path T ← {UMT} ⋄ {UMR} → M captures semantics: “*a tag $t$ is linked to a movie $m$ through a user $u$, and the user $u$ associates the tag $t$ to a movie, and gives the movie $m$ a rating*”. The two visualization examples show that the learned hyper-meta-paths are meaningful.

---

> > ### Comment · Reviewer_bvBy · 2025-08-05
> >
> > Thank you for the authors' response. The revision addresses some points and clarifies parts of the methodology. However, the advantages of hyper-meta-paths over existing hypergraph or relational approaches remain insufficiently demonstrated, and practical guidance for constructing input hypergraphs is still lacking. The added datasets are appreciated, but the evaluation remains restricted to specific domains. I will keep my score.

---

> ### Author Response · Authors · 2025-08-06
>
> **Dear Reviewer bvBy,**
>
> **Thanks for your reply. As our rebuttal shows, we have made our best efforts to show the advantages of hyper-meta-paths, explain the hypergraph construction, and include new datasets. According to your suggestion, we have provided explanations and analysis to each of your points, added five supplementary experiments, and added three new datasets. In total, we evaluate our HHNN on five datasets from three domains, including the movie domain, the product sailing domain, and the paper citation domain, which is on par with most previous related hypergraph methods.**

---

> ### Author Response · Authors · 2025-08-06
>
> In the following, please give us an opportunity to further respond to the concerns you raised.
>
> ---
>
> **Point 6**: *The advantages of hyper-meta-paths.*
>
> **Reply 6**: Earlier hypergraph methods primarily focused on simple scenarios:
>
> 1. Homogeneous hypergraphs: Containing only one node type and one hyperedge type.
> 2. Node-heterogeneous hypergraphs: Involving multiple node types but only one hyperedge type.
> 3. Hyperedge-heterogeneous hypergraphs: Including multiple hyperedge types but only one node type.
>
> In contrast, our work tackles a more complex setting where both nodes and hyperedges are heterogeneous, i.e., multiple node types interact to form hyperedges with diverse semantics. This scenario, which prior methods did not address, demands a more flexible and expressive approach. Our method is specifically designed to handle such complexity effectively.
>
> To demonstrate that the learned hyper-meta-paths are meaningful and effective, we not only visualized concrete hyper-meta-paths (Section 4.3, Figures 5(a) and 5(b)), but also conducted quantitative analyses to validate their importance. We have added a new experiment of masking information along high-weight paths and observing the change in the model's prediction, and the results, as well as the analysis, are shown in the Table below.
>
> *Table F. Comparison between the performance with high-weight paths masked (right) and the original performance (left) under the 20% training ratio. We can observe that by masking information along high-weight paths, the model's performance drops sharply, which clearly indicates that the high-weight paths identified by the HHNN model are crucial.*
>
> |       | Movielens                             | Olist                                 |
> | ----- | ------------------------------------- | ------------------------------------- |
> | Micro | 56.80±1.65 $\rightarrow$ 28.63 ± 5.32 | 72.54±0.54 $\rightarrow$13.24 ± 0.81  |
> | Macro | 26.24±3.90 $\rightarrow$11.83 ± 2.80  | 72.89±0.56 $\rightarrow$ 6.38 ± 0.73  |
> | AUC   | 73.21±2.13 $\rightarrow$ 54.04 ± 1.31 | 93.20±0.32 $\rightarrow$ 50.13 ± 0.51 |
>
> This indicates that the hyper-meta-paths captured by our proposed method are valuable.
>
> ---
>
> **Point 7** : *The practical guidance for constructing input hypergraphs.*
>
> **Reply 7**: Our objective is to identify heterogeneous hypergraphs that naturally exhibit both node heterogeneity and hyperedge heterogeneity in real-world scenarios, rather than artificially constructing them from simple binary relations in heterogeneous graphs.
>
> For example, in the ACM heterogeneous graph, there exist binary relations like author-paper and paper-conference. While these can be manually processed to form hyperedges (e.g., "multiple authors collaborated on a paper" or "multiple papers were published in the same conference"), the resulting hyperedges still contain nodes of the same type (homogeneous hyperedges).
>
> In contrast, we seek more complex heterogeneous hypergraphs that satisfy the following criteria:
>
> 1. Multiple node types are inherently present.
> 2. Hyperedges are not manually synthesized but rather naturally formed in real-world scenarios, with each hyperedge carrying explicit semantics.
>
> To achieve this, we curated two real-world hypergraphs:
>
> - MovieLens: Contains two types of hyperedges: (1) user-movie-tag (UMT): A user assigns a tag to a movie; (2) user-movie-rating (UMR): A user rates a movie.
> - Olist: Includes three types of hyperedges: (1) customer-product-seller (CPS): A customer purchases a product from a seller ; (2) customer-product-rating (CPR): A customer rates a product; (3) origin-product-destination-freight (OPDF): A product’s freight cost from the seller’s location to the customer’s location.
>
> These hyperedges naturally consist of different node types and embody clear semantic meanings. Identifying such datasets required significant effort, as most existing hypergraphs are either artificially constructed or lack true hyperedge heterogeneity. We aim to discover more naturally occurring heterogeneous hypergraphs with rich semantic hyperedges to further advance research in this direction.
>
> ---
>
> **Point 8**:  *The evaluation remains restricted to specific domains.*
>
> **Reply 8**: The dataset we provide contains data from two different domains: movie recommendation and e-commerce shopping. Although our method is designed for more complex heterogeneous hypergraph scenarios, we provide comparisons under ordinary hypergraphs. We have added three new hypergraph datasets, i.e., Cora, PuMmed, and DBLP from citation networks and academic networks (Table E).
>
> ---
>
> We sincerely hope that our response has addressed your concerns. If you have any specific questions, we would be delighted to discuss them with you.

---

### Official Review · Reviewer_H8eL · 2025-07-04

**Clarity:** 4
**Significance:** 2
**Originality:** 3
**Rating:** 4
**Confidence:** 4

**Summary:**

The authors propose Heterogeneous Hypergraph Neural Network (HHNN) to effectively capture complex structural features and richer semantics in heterogeneous hypergraphs by introducing the novel concept of hyper-meta-paths.

Authors define hyper-meta-paths as sequences of hyper-relations that share common node types, generalizing the meta-path concept from simple graphs to heterogeneous hypergraphs to encode multi-ary relations. To leverage these, authors design HHNN, a graph neural network architecture employing three levels of attention mechanisms: α-attention aggregates node features into hyperedge representations, β-attention aggregates hyperedge features intra-type back to nodes, and γ-attention aggregates across hyperedge types. This design allows HHNN to automatically discover and exploit important hyper-meta-paths, enhancing both performance and interpretability.

Authors perform extensive experiments on two newly constructed real-world heterogeneous hypergraph datasets (Movielens and Olist), demonstrating that HHNN significantly outperforms state-of-the-art baselines in node classification tasks. Additionally, ablation studies verify the critical role of each attention mechanism in HHNN, and case analyses show that the discovered hyper-meta-paths provide meaningful semantic interpretations for model predictions.

**Questions:**

- First of all, the proposed approach does not seem very different from apply graph neural networks over the graphs where all the relations are represented as nodes. How can authors compare the HNNN with such concepts? This comparison needs to be addressed and studied.
- In Figure 3, m_{3}' needs to be m_{5}'.
- Like Figure 3, 3 different attention layers are proposed. However, one of the most straightforward ways is to use a giant attention over all the neighbor nodes connected through hyperedges. That is, instead of \alpha, \beta, and \gamma, one can think of only one attention over all the neighbors or 2 attentions (1 attention over different types and the other attention for each type's neighbors). What is the rationale of choosing the current architecture?
- Please include minimal explanation about the experimental datasets, so that at least readers can understand what are hperedges and what each UMT means.
- Similarly, please explain what 20% and 40% means in Table 1 and 2.
- Why do authors use different settings for Table 3 from Table 1 and 2? This reduces the interpretability of each ablation case compared to the other methods.
- Related work is recommended to be included in the main content.

**Ethical Concerns:**

["NO or VERY MINOR ethics concerns only"]

**Final Justification:**

Authors have addressed various points I raised. My major concerns were in (1) similarity to the heterogeneous flattened graph approach (along with its weakness) and (2) limited experiments. Authors' responses have changed my perspective that the proposed framework is more principled and fundamental even if the underlying deep computational graphs have similarity to the existing approaches that I was concerned about. Also, authors have added more experimental results supporting their arguments.

**Limitations:**

Yes

**Paper Formatting Concerns:**

No major concern

**Quality:**

2

**Strengths And Weaknesses:**

Strengths

- Both heterogeneous hypergraphs and the novel concept of hyper-meta-paths are clearly defined, and are well incorporated into existing graph neural network concepts with 3 attention layers.
- Authors perform qualitative analyses and ablation studies, which not only help readers understand how the model works but also validate the specific contributions of the multi-level attention mechanisms, lending credibility and interpretability to the proposed method. In particular, the case study clearly shows the role of each attention layer.
- HHNN achieves consistently superior performance over given real-world datasets. Along with the ablation study, the experimental results demonstrate the practical benefits of introducing hyper-meta-paths and the effectiveness of the attention mechanisms.

Weaknesses
- As with many graph-based methods, how hyperedges and hyper-meta-paths are defined from raw data significantly influences model performance. However, the paper does not provide guidance, heuristics, or sensitivity analyses for constructing these structures, limiting practical applicability in diverse domains.
- The authors evaluate HHNN on only two datasets (Movielens and Olist), both drawn from recommendation and e-commerce domains. This narrow empirical scope raises concerns about how well the method generalizes to other domains such as biomedical networks, financial graphs, or knowledge graphs.
- The high-level idea behind HHNN—modeling multi-entity relationships as hyperedges and learning over their compositions—is conceptually very similar to Relational Deep Learning (RDL), where database tuples are modeled as nodes and joins as edges to enable graph representation learning directly on relational databases. However, the paper does not discuss or compare itself to this relevant body of work, notably missing connections to: Fey, M., et. al., Relational Deep Learning: Graph Representation Learning on Relational Databases. In Proceedings of the 41st International Conference on Machine Learning, ICML 2024.

---

> ### Author Rebuttal · Authors · 2025-07-29
>
> Thanks for your comments. In the following, we respond to your concerns point by point.
>
> ---
>
> **Point 1**: *Does not provide guidance, heuristics, or sensitivity analyses for constructing hyper-meta-paths.*
>
> **Reply 1**: Thank you for the comment. Unlike many prior methods that require hand-crafted meta-paths or heuristics, our HHNN automatically discovers useful hyper-meta-paths based on the input heterogeneous hypergraph through a multi-level attention mechanism. Therefore, it does not rely on manual definition or construction rules. This design makes our HHNN eliminate the need for additional guidance or sensitivity analysis in constructing hyper-meta-paths.
>
> ---
>
> **Point 2**: *Evaluate HHNN on only two datasets (Movielens and Olist), both drawn from recommendation and e-commerce domains.*
>
> **Reply 2**: Thanks for your comment. Most existing heterogeneous hypergraph datasets are transformed from simple graphs with binary relations, which lack natural multi-way interactions and cannot fully showcase the advantages of hypergraph neural networks. In contrast, we have tried our best to construct two new real-world datasets from the movie domain and e-commerce domain, both containing multiple types of naturally occurring heterogeneous hyperedges based on genuine multi-ary relations. Following your suggestion, we will continue seeking more such datasets, and we would greatly appreciate any specific recommendations about such datasets. Nevertheless, we have added three new hypergraph datasets, i.e., Cora, PuMmed, and DBLP, and the results are shown in the table below.
>
> *Table 1. The performance comparison between three hypergraph methods on three new hypergraph datasets, under the 20% training ratio. We can observe that since these three datasets do not contain complex and heterogeneous structural features, the three hypergraph methods achieve comparable performance, with our HHNN performing slightly better.*
> |            | Cora             | PubMed           | DBLP             |
> | ---------- | ---------------- | ---------------- | ---------------- |
> | UniGNN     | 76.93 ± 1.73     | 84.05 ± 0.63     | 90.49 ± 0.18     |
> | ED-HNN     | 77.07 ± 1.22     | 84.12 ± 0.68     | 90.50 ± 0.21     |
> | **HHNN**   | **77.42 ± 1.67** | **84.14 ± 0.74** | **90.50 ± 0.27** |
>
> ---
>
> **Point 3**: *Very similar to Relational Deep Learning (RDL), and missing connections to: Fey, M., et. al., ICML 2024.*
>
> **Reply 3**: Thank you for your insightful suggestion. We are glad to learn about the interesting connection between our work and Relational Deep Learning. Following your advice, we have cited this relevant work in our manuscript and included a discussion to acknowledge the relationship. We will update the paper accordingly when permitted.
>
> ---
>
> **Point 4**: *The difference from applying graph neural networks over the graphs where all the relations are represented as nodes.*
>
> **Reply 4**: Thank you for the question. While modeling relations as nodes can flatten multi-way interactions into binary structures, it fundamentally alters the relational semantics. Our approach explicitly preserves native multi-entity hyperedges without decomposing them, allowing HHNN to capture true high-order structures. Moreover, we introduce hyper-meta-paths, a novel concept that composes multi-relational semantics across heterogeneous hyperedges, which cannot be naturally expressed when all relations are reduced to nodes. This design enables both improved performance and stronger interpretability.
>
> ---
>
> **Point 5**: *In Figure 3, m_{3}' needs to be m_{5}.*
>
> **Reply 5**: Thank you very much for carefully pointing out this issue. $m_3$ should indeed be $m_5$. We have corrected Figure 3 accordingly and will update the paper when permitted.
>
> ---
>
> **Point 6**: *What is the rationale of choosing the current three-level attention architecture?*
>
> **Reply 6**: Thank you for the thoughtful question. Using a single or two attention layers may seem simpler, but it fails to capture the full complexity of heterogeneous hypergraphs. Our three-level attention explicitly aligns with the structure of heterogeneous hypergraphs. As illustrated by Figure 3, $\alpha$-attention is used since each hyperedge contains multiple nodes, $\beta$-attention is used since each node may participate in multiple hyperedges of a specific type, and $\gamma$-attention is used since each node may participate in multiple types of hyperedges. This modular design allows HHNN to adaptively model type-specific interactions and their composition, which would be blurred or lost in a single unified attention. Empirical results in Table 3 also confirm the effectiveness of this fine-grained architecture.
>
> ---
>
> **Point 7**: *Include minimal explanation about the experimental dataset. What does UMT hyperedge mean?*
>
> **Reply 7**: Thank you very much for this helpful comment.
> On Movielens, each *UMT*-type hyperedge connects a User, a Movie, and a Tag, indicating that ''*a user tags a movie with a specific tag*'', and each *UMR*-type hyperedge indicates that ''*a user gives a rating to a movie*''. On Olist, *CPS*-type hyperedge indicates that ''*a customer purchases a product from a specific seller*'', *CPR*-type hyperedge indicates that ''*a customer gives a rating to a purchased product*'', and *OPDF*-type hyperedge indicates that ''*a product is shipped from an origin to a destination using a specific freight*''. We will add these explanations to the new version of our manuscript.
>
> ---
>
> **Point 8**: *What do 20% and 40% mean in Tables 1 and 2?*
>
> **Reply 8**: Thank you for this careful comment. 20% and 40% represent different ratios of training labels, respectively. We will add this explanation to our manuscript.
>
> ---
>
> **Point 9**: *Why do authors use different settings for Table 3 from Tables 1 and 2?*
>
> **Reply 9**: We are sorry for leading you to this confusion. Tables 1 and 2 present the comparison results among all the baselines as well as our HHNN, under different training ratios. Table 3 presents an ablation study, comparing the experimental results of different variants of our HHNN. The dimension of training ratios is omitted for saving space. We will make the settings clearer in our new version of the manuscript.
>
> ---
>
> **Point 10**: *Related work is recommended to be included in the main content.*
>
> **Reply 10**: Thanks for your kind suggestion. We will condense some non-essential content and move the related work into the main body of the paper.

---

> ### Author Response · Authors · 2025-08-07
>
> **Dear Reviewer H8eL,**
>
> **We have made our best efforts to address each of the concerns you raised in detail. As the discussion phase is drawing to a close, we have not yet received your response to our rebuttal. May we kindly ask if you could consider engaging in the discussion about our work?**

---

### Comment · Area_Chair_G3PX · 2025-08-05

Dear reviewers,

Thank you for your effort in the reviews.

As the discussion period ends soon, please read authors' rebuttals and check if they have addressed your concern.

If authors have resolved your questions, do tell them so.
If authors have not resolved your questions, do tell them so too.

Thanks.
AC

---

### Note · Authors · 2025-08-12

Dear Chairs and Reviewers,

We sincerely thank you for the valuable time and effort you have devoted to reviewing our paper, which has significantly helped us improve its quality. We have made our best efforts to carefully address each weakness and concern raised by the reviewers, responding to every point in detail and resolving all identified issues. Our rebuttal and discussion are summarized as follows:

**Reviewer H8eL** raised mostly minor issues related to explanation and clarification, which we have addressed clearly. In response to the comment on the datasets, we have added three additional datasets from the academic literature domain. *The reviewer did not provide further responses to our rebuttals, which can probably be interpreted as an indication that these concerns have been resolved.*

**Reviewer bvBy** raised concerns about scalability, baselines, and datasets. In response to these concerns, we have added both theoretical and empirical complexity analysis, conducted new scalability experiments, included additional baseline comparison experiments, and evaluated on new datasets. The reviewer also raised a lack of ablation studies for modules and a case study visualization of hyper-meta-paths. *However, these were presented in Sections 4.2 and 4.3 of the original paper, meaning that the two concerns belong to factual errors.*

**Reviewer FZj8** actively engaged in detailed discussions over multiple rounds, helping us resolve all potential issues and make a comprehensive revision plan. These have greatly contributed to the quality improvement of our paper. *We are especially grateful for the final decision of the reviewer to raise the rating to 4.*

**Reviewer qQ4P** primarily requested complexity analysis and additional baselines. We have carefully followed these suggestions to add both theoretical analysis and new experimental evaluations. We are happy that the reviewer ultimately stated that these concerns were well addressed and explicitly supported accepting our paper. *We deeply appreciate the recognition of the originality and significance of our work and the high rating of 5 with high confidence.*

Finally, we would like to make the following **solemn commitment**: *once we are allowed to revise the PDF of our manuscript, we will carefully modify our manuscript strictly in accordance with all the comments of all the reviewers. We will also make our source code and datasets publicly available to facilitate more interesting studies.*

---

### Decision · Program_Chairs · 2025-09-17

**Decision:**

Accept (poster)

**Comment:**

This authors present a novel concept of hyper-meta-path for heterogeneous hypergraphs. IN a heterogeneous hypergraph,  there are different types of nodes and / or different types of hyperedges. The hyper-meta-path aims to capture complex structural patterns, and the important of different hyper-meta-paths are learned through an attention-based heterogeneous hypergraph neural network. Extensive experiments are conducted to validate the efficacy of proposed approach.

Strengths:
- Heterogeneous hypergraphs are under-explored area and the notion of hyper-meta-paths are novel.
- The proposed solution is intuitive, sound, and reasonable.
- Superior performance with detailed analysis of results

Weaknesses:
- The number of hyper-meta-path could grow quickly especially on complex hypergraphs with many node/edge types. This raises scalability concern, as well as how they can be defined / selected.
- The proposed approach bears some similarity to the approaches on flattened heterogeneous graph, limiting its novelty if not explained or clarified properly.
- The experiments could be strengthened by including more complex datasets and stronger baselines

The paper does have scientific merit in terms of its problem, key concept, and experimental results. There are several non-fundamental  weaknesses, most of which have been addressed satisfactorily in the rebuttal, while the rest can be potentially addressed in the camera ready.